# Antibiotic Dispensation without a Prescription Worldwide: A Systematic Review

**DOI:** 10.3390/antibiotics9110786

**Published:** 2020-11-07

**Authors:** Ana Daniela Batista, Daniela A. Rodrigues, Adolfo Figueiras, Maruxa Zapata-Cachafeiro, Fátima Roque, Maria Teresa Herdeiro

**Affiliations:** 1Department of Medical Sciences, University of Aveiro, 3810-193 Aveiro, Portugal; anadbaptista@ua.pt; 2Research Unit for Inland Development, Polytechnic Institute of Guarda (UDI/IPG), 6300-559 Guarda, Portugal; danielaalmeidar@ipg.pt (D.A.R.); froque@ipg.pt (F.R.); 3Department of Preventive Medicine and Public Health, University of Santiago de Compostela, 15702 Santiago de Compostela, Spain; adolfo.figueiras@usc.es (A.F.); maruxa.zapata@usc.es (M.Z.-C.); 4Consortium for Biomedical Research in Epidemiology and Public Health (CIBER Epidemiology and Public Health-CIBERESP), 28001 Madrid, Spain; 5Health Research Institute of Santiago de Compostela (IDIS), 15702 Santiago de Compostela, Spain; 6Health Sciences Research Centre, University of Beira Interior (CICS-UBI), 6200-506 Covilhã, Portugal; 7Institute of Biomedicine (iBiMED), Department of Medical Sciences, University of Aveiro, 3810-193 Aveiro, Portugal

**Keywords:** antibiotics, non-prescription antibiotic dispensing, pharmacy practice, self-medication

## Abstract

Antibiotic resistance still remains a major global public health problem and the dispensing of antibiotics without a prescription at community pharmacies is an important driver of this. MEDLINE, Pubmed and EMBASE databases were used to search and identify studies reporting the dispensing of non-prescribed antibiotics in community pharmacies or drugstores that sell drugs for human use, by applying pharmacy interviews/questionnaires methods and/or simulated patient methods. Of the 4683 studies retrieved, 85 were included, of which 59 (69.4%) were published in low-and middle-income countries. Most of the papers (83.3%) presented a percentage of antibiotic dispensing without a prescription above 60.0%. Sixty-one studies evaluated the active substance and the most sold antibiotics without a prescription were amoxicillin (86.9%), azithromycin (39.3%), ciprofloxacin (39.3%), and amoxicillin-clavulanic acid (39.3%). Among the 65 articles referencing the diseases/symptoms, this practice was shown to be mostly associated with respiratory system problems (100.0%), diarrhea (40.0%), and Urinary Tract Infections (30.8%). In sum, antibiotics are frequently dispensed without a prescription in many countries and can thus have an important impact on the development of resistance at a global level. Our results indicate the high need to implement educational and/or regulatory/administrative strategies in most countries, aiming to reduce this practice.

## 1. Introduction

Antibiotic resistance remains one of the major global public health problems, due to its impact on morbidity, mortality, and healthcare costs. Despite the alerts from international organizations [1,2,3], the trend towards antibiotic consumption is continuing to increase in some countries, especially in low-and middle-income countries [4]. In recent years, some high-income countries have manged to decrease the consumption of antibiotics, suggesting that the educational/regulatory strategies developed over the past few years are having some impact on slowing down the consumption [5] projections of global antibiotic consumption in 2030, assuming no policy changes, by up to 200% [4].

Although many factors can explain these increases in antibiotic use and resistance, the dispensing of antibiotics without prescription in community pharmacies is still one of the most important factors [6,7]. It is estimated that over-the-counter dispensing may account for half of the antibiotic sales worldwide [8], even though in North America, Northern Europe, and Australia this pratically doe not exist [7]. This self-medication is associated with an inadequate use of antibiotics, inadequate dosages, and duration of treatments that increase the risk of resistant bacteria selection [9].

To assess the magnitude of dispensing antibiotics without prescription, several methods have been proposed, mainly in simulated patients and pharmacy interviews, each with its advantages and limitations. On the one hand, simulated patients (an individual visiting a pharmacy simulating specific symptoms and requiring an antibiotic) avoid the Hawthorne effect (changes in the behavior of the studied pharmacists because they feel observed), and make it impossible to speak about complacency, since these patients are generally unknown to the pharmacist [10]. In contrast, pharmacy interviews allow a qualitative analysis and the possibility of introducing open questions, thus displaying a greater Hawthorne effect and an evaluation of antibiotics dispensing due to complacency. Therefore, it seems that both methods can provide complementary information.

This systematic review aims to identify and compare the frequency of dispensing antibiotics without a prescription in the community pharmacies or drugstores that sell drugs for human use, by using two different methods, pharmacy interviews/questionnaires and simulated patients.

## 2. Results

### 2.1. Search Results

The chosen search strategy identified a total of 4871 articles in MEDLINE Pubmed and Embase databases, being that 188 articles were duplicated. After screening for title and abstract, 188 articles potentially met the inclusion criteria and were selected for a reading of the complete text. Following an in-depth reading, a total of 85 papers were finally included in this systematic review (Figure 1).

### 2.2. Quality Assessment

All the studies included in the systematic review complied with most of the questions in the AXIS tool (Table 1). A high proportion of articles (64/85) fulfilled almost all of the criteria and were thus regarded as displaying a high methodological quality and a low susceptibility to bias. Nineteen articles failed to comply with four to six of the exploratory questions and were considered to have a medium level of methodological quality. Two of the remaining articles failed to comply with more than 6 exploratory questions and were classified as having a low level of methodological quality.

### 2.3. Characteristics of the Studies Selected

A description of the general characteristics of the 85 included studies is presented in Table 1.

Among the included studies, 51 were conducted in Asia, 17 in Africa (including Egypt), 14 in Europe (including Turkey), 2 in South America, and 1 in North America.

Regarding the study design, almost all of the included papers were cross-sectional studies (72 of 85), 7 were prospective studies, 3 were cross-sectional prospective studies, 1 was a qualitative exploratory study, 1 was a quantitative study, and 1 was an interventional study.

Of the total of 85 studies included in this review, 49 used the simulated patient method (Table 2) as a data collection method and 33 used the pharmacy interviews/questionnaires method (Table 3). Three of the studies included in the review used both methods.

### 2.4. Study Outcomes—Articles that Use the Simulated Patient Method

Table 2 summarizes the extent of antibiotic dispensation without prescription and additional outcomes of the included articles that use the simulated patient method.

#### 2.4.1. Frequency of Antibiotic Dispensation without a Prescription

Among the 52 articles that used a simulated patient method, 6 of them did not explicitly mention throughout the full article that it was about dispensing antibiotics without a prescription, but rather the management of diseases/symptoms in pharmacies/drug stores [11,12,13,14,15,16]. The inclusion of these articles is justified by the fact that they present data related to the dispensing of antibiotics, without mentioning the presence of a prescription anywhere in the article.

The frequency of antibiotic dispensation without a prescription is presented to (i) the total number of pharmacies/drugstores/medicine retail outlets visited (40 articles); (ii) total number of visits made (9 articles); (iii) total number of interactions (2 articles); and finally about (iv) the total number of drugs dispensed (1 article).

Among the 40 articles that were presented, the dispensation of antibiotics to the total number of pharmacies/drug stores/medicine retail outlets, the percentage of dispensation without a prescription of these establishments, varied from 8.0% to 100.0%. About 25 articles made a percentages of dispensation without a prescription above 60.0%.

Of the 9 articles that presented the percentage of dispensation without a prescription regarding the total number of visits made, the percentage ranged from 5.0% to 87.6% of visits, resulting in an antibiotic being dispensed without a prescription.

The two articles that presented the percentage of dismissal without a prescription in relation to the total number of interactions varied from 4.0% to about 59.3%. Finally, only the article by Ibrahim M.I.B. et al. [13] presented the percentage of dispensation without a prescription concerning the total number of medicines dispensed, and obtained a value of 43.2%.

Those by Al-Tannir M. et al. [17], held in Saudi Arabia, and Marković-Peković V et al. [18], held in Republic of Srpska and Herzegovina, compared the frequency of dispensing antibiotics without prescription in two different years, 2011 and 2018 and 2010 and 2015, respectively. In both articles, it is possible to observe a decrease in the frequency of dispensing antibiotics without a prescription from the past to the most recent year (Al-Tannir M. et al. [17] (2011: 77.6%; 2018: 12.5%); Marković-Peković V et al. [18] (2010: 58.0%; 2015: 18.5%)).

The highest percentages of antibiotic dispensation without a prescription associated with the simulated patient method corresponded to studies carried out in Asian countries. However, the lowest percentage of antibiotic dispensation without a prescription was recorded in a study conducted in India [19].

Three articles conducted in Tanzania and Thailand [20,21,22] presented the percentage of antibiotic dispensing without a prescription related to class I, legally authorized to dispense antibiotics without prescription and class II that only sell them with a medical prescription. Two [20,22] of these articles presented the percentage related to the two pharmacy classes, while 1 article [21] only presented the percentage related to class I pharmacies. By comparing the percentages presented linked to both types of pharmacy, it is possible to verify that a higher percentage is related to the type II pharmacies in the two articles presented.

Additionally, the article by Minzi O et al. [23], held in Tanzania, also presents the percentage of antibiotic dispensation without a prescription related to the accredited drug dispensing outlets (ADDO) and the duka la dowa baridi (DLDB) also know as class II shops. According to the regulations in Tanzania, DLDB are not authorised to sell prescription drugs, such as antibiotics. In contrast, ADDO have been developed to improve the availability of medicines and are authorised to sell a limited range of prescription medicines, among which are a restricted number of antibiotics. [23] In this article, the authors reported a higher percentage related to ADDO, when compared to DLDB.

#### 2.4.2. Name/Class of Antibiotics Most Often Dispensed without a Prescription

There were several antibiotics dispensed without a prescription. The antibiotics most commonly mentioned in the included articles were amoxicillin (40 articles), azithromycin (24 articles), ciprofloxacin (24 articles), and amoxicillin-clavulanic acid (21 articles).

#### 2.4.3. Types of Disease/Symptoms Most Commonly Associated with Dispensation without a Prescription

The most used diseases/symptoms in the simulated patient methodology were diarrhea (23 articles), urinary tract infections (UTI) (12 articles), upper respiratory tract infections (URTI) (11 articles), sore throat (11 articles), common cold (6 articles), and bronchitis (6 articles). Others included cough (5 articles), fever (5 articles), sinusitis (3 articles), otitis media (3 articles), rhinitis (3 articles), and gastroenteritis (2 articles).

#### 2.4.4. Level of Insistence by Patients

Of the 52 articles included in the systematic review, 21 used different levels of demand to obtain antibiotics. Of these, 15 studies applied 3 levels of demand, 4 studies used 2 levels and, finally, 2 studies presented 4 levels.

From the results obtained, it was possible to verify that most of the articles that presented 3 levels of demand indicated that the majority of antibiotics were obtained at level 1, meaning they were only obtained when something was asked for to alleviate the symptoms (6 articles).

In contrast, of the 4 articles that used 2 levels of demand, most indicated that antibiotics were obtained according to level 2, that is when asking for a stronger medication or, more specifically, an antimicrobial.

Of the 2 articles that used 4 levels of demand, only 1 presented information about the level at which the highest amount of antibiotics was obtained, namely level 4, corresponding to the specific request of the amoxicillin antibiotic.

**Table 1 antibiotics-09-00786-t001:** Characteristics of the selected studies.

Authors	Year of Publication	Country	Study Design	Data Collection Method	Quality Assesment ^a^	Criteria not Fulfilled ^a^
Abubakar U. et al. [24]	2020	Nigeria	Cross-sectional prospective study	Pharmacy interviews/questionnaires	High	7
Abubakar U. [25]	2020	Nigeria	Cross-sectional study	Pharmacy interviews/questionnaires	High	6, 7, 13
Al-Tannir M. et al. [17]	2020	Saudi Arabia	Cross-sectional study	Simulated patient method	High	7, 12, 14
Badro D.A. et al. [26]	2020	Lebanon	Cross-sectional study	Pharmacy interviews/questionnaires	High	3, 7, 14
Bahta M. et al. [27]	2020	Eritrea	Cross-sectional study	Simulated patient method	High	7, 13, 14
Chen J. et al. [28]	2020	China	Cross-sectional study	Simulated patient method	High	6, 7, 14
Gajdács M. et al. [29]	2020	Hungary	Cross-sectional study	Pharmacy interviews/questionnaires	High	3, 7, 14
Halboup A. et al. [30]	2020	Yemen	Cross-sectional study	Simulated patient method	High	14
Shi L. et al. [31]	2020	China	Cross-sectional study	Simulated patient method	High	6, 7, 14
Wang X. et al. [32]	2020	China	Cross-sectional study	Simulated patient method	High	12, 14
Abdelaziz AI et al. [33]	2019	Egypt	Cross-sectional study	Simulated patient method	High	14
Alrasheedy AA. et al. [34]	2019	Saudi Arabia	Cross-sectional study	Simulated patient method and pharmacy interviews/qestionnaires	Medium	6, 7, 12, 14
Chang J. et al. [7]	2019	China	Cross-sectional study	Simulated patient method	High	6, 7, 14
Damisie G et al. [35]	2019	Ethiopia	Cross-sectional study	Simulated patient method	Medium	3, 7, 10, 14
Hallit S et al. [36]	2019	Lebanon	Cross-sectional study	Pharmacy interviews/questionnaires	High	7, 13
Koji EM et al. [37]	2019	Ethiopia	Cross-sectional prospective study	Simulated patient method	High	7, 14
Mengistu G et al. [11]	2019	Ethiopia	Cross-sectional study	Simulated patient method and pharmacy interviews/questionnaires	High	7, 14
Nafade V. et al. [19]	2019	India	Cross-sectional study	Simulated patient method	High	7, 14
Zawahir S et al. [38]	2019	Sri Lanka	Cross-sectional study	Simulated patient method	High	7, 14
Zawahir S. et al. [39]	2019	Sri Lanka	Cross-sectional study	Pharmacy interviews/questionnaires	High	14
Ajie A.A.D. et al. [40]	2018	Indonesia	Cross-sectional study	Pharmacy interviews/questionnaires	High	18
Alhomoud F et al. [41]	2018	Saudi Arabia	Qualitative exploratory study	Pharmacy interviews/questionnaires	Medium	6, 7, 13, 14
Awosan KJ et al. [42]	2018	Nigeria	Cross-sectional study	Pharmacy interviews/questionnaires	High	13, 14
Erku D.A. et al. [43]	2018	Ethiopia	Cross-sectional study	Simulated patient method	High	3, 7, 14
Horumpende PG et al. [20]	2018	Tanzania	Cross-sectional study	Simulated patient method	High	7, 14
Ibrahim IR et al. [12]	2018	Iraq	Cross-sectional study	Simulated patient method	High	6, 7, 14
Mohamed Ibrahim M.I. et al. [44]	2018	Qatar	Cross-sectional study	Simulated patient method	High	7, 14
Paes M.R. et al. [45]	2018	India	Cross-sectional study	Pharmacy interviews/questionnaires	Medium	3, 7, 10, 11, 12, 14
Rehman IU et al. [46]	2018	Pakistan	Cross-sectional study	Pharmacy interviews/questionnaires	Medium	3, 7, 13, 14
Sarwar M.R. et al. [47]	2018	Pakistan	Cross-sectional study	Pharmacy interviews/questionnaires	High	7, 14
Zapata-Cachafeiro M et al. [48]	2018	Spain	Cross-sectional study	Simulated patient method	High	7, 14
Zawahir S et al. [49]	2018	Sri Lanka	Cross-sectional study	Simulated patient method	High	14
Ansari M. [50]	2017	Nepal	Cross-sectional prospective study	Pharmacy interviews/questionnaires	High	7, 14
Barker A.K. et al. [51]	2017	India	Cross-sectional study	Pharmacy interviews/questionnaires	High	7, 14
Chang J. et al. [52]	2017	China	Cross-sectional study	Simulated patient method	High	7, 14
Jaisue S. et al. [21]	2017	Thailand	Cross-sectional study	Simulated patient method	Medium	3, 7, 10, 14, 18
Mansour O. et al. [53]	2017	Syria	Cross-sectional study	Pharmacy interviews/questionnaires	High	7, 13
Marković-Peković V et al. [18]	2017	Republic of Srpska and Herzegovina	Cross-sectional study	Simulated patient method	High	7, 14, 18
Okuyan B. et al. [54]	2017	Turkey	Cross-sectional study	Simulated patient method	High	7, 14, 18
Abegaz T.M. et al. [14]	2016	Ethiopia	Cross-sectional study	Simulated patient method	High	6, 7, 14
Abood EA et al. [55]	2016	Yemen	Cross-sectional study	Pharmacy interviews/questionnaires	High	3, 7, 14
Erku D.A. [56]	2016	Ethiopia	Cross-sectional study	Pharmacy interviews/questionnaires	High	7, 14
Guinovart MC et al. [57]	2016	Spain	Prospective study	Simulated patient method	Medium	3, 7, 10, 14, 18, 20
Ibrahim M.I.B.M. et al. [13]	2016	Qatar	Cross-sectional study	Simulated patient method	High	7, 14
Kalungia AC et al. [58]	2016	Zambia	Descriptive cross-sectional study	Pharmacy interviews/questionnaires	High	7, 14
Khan M.U. et al. [59]	2016	Malaysia	Cross-sectional study	Pharmacy interviews/questionnaires	High	7, 13, 14
Nawab A. et al. [60]	2016	Pakistan	Cross-sectional study	Pharmacy interviews/questionnaires	Medium	6, 7, 10, 12, 14, 18
Satyanarayana S. et al. [61]	2016	India	Cross-sectional study	Simulated patient method	High	6, 7, 14
Almaaytah A et al. [62]	2015	Jordan	Prospective study	Simulated patient method	Medium	7, 10, 14, 18
Bahnassi A. [63]	2015	Syria	Cross-sectional study	Pharmacy interviews/questionnaires	High	7, 10, 18
Dorj G. et al. [64]	2015	Mongolia	Cross-sectional study	Pharmacy interviews/questionnaires	High	7, 14
Shet A et al. [65]	2015	India	Cross-sectional study	Simulated patient method	High	7, 14
Shreya Svitlana A. et al. [66]	2015	India	Cross-sectional study	Pharmacy interviews/questionnaires	Low	6, 7, 9, 10, 11, 12, 14, 18, 20
Alabid A.H.M.A. et al. [67]	2014	Malaysia	Cross-sectional study	Simulated patient method	High	7, 14
Bahnassi A. [68]	2014	Saudi Arabia	Cross-sectional study	Pharmacy interviews/questionnaires	High	7, 14
Farah R et al. [69]	2014	Lebanon	Cross-sectional study	Pharmacy interviews/questionnaires	High	7, 14, 20
Gastelurrutia M.A. et al. [70]	2014	Spain	Prospective study	Pharmacy interviews/questionnaires	Medium	3, 7, 10, 11, 14
Sabry NA et al. [71]	2014	Egypt	Cross-sectional study	Pharmacy interviews/questionnaires	High	7
Zapata-Cachafeiro M. et al. [72]	2014	Spain	Cross-sectional study	Pharmacy interviews/questionnaires	High	7, 9, 14
Abasaeed AE et al. [73]	2013	United Arab Emirates	Cross-sectional study	Pharmacy interviews/questionnaires	High	7, 14
Malik M. et al. [15]	2013	Pakistan	Cross-sectional study	Simulated patient method	High	3, 7, 18
Minzi O et al. [23]	2013	Tanzania	Cross-sectional study	Simulated patient method	High	7, 9, 14
Marković-Peković V et al. al. [74]	2012	Republic of Srpska and Herzegovina	Cross-sectional study	Simulated patient method	Medium	3, 7, 10, 14, 20
Rathnakar U.P. et al. [75]	2012	India	Prospective study	Simulated patient method	High	7, 14, 18
Simó S et al. [76]	2012	Spain	Prospective study	Simulated patient method	Medium	7, 12, 14, 20
Al-Faham Z et al. [77]	2011	Syria	Cross-sectional study	Simulated patient method	High	7, 10, 20
Al-Mohamadi A et al. [78]	2011	Saudi Arabia	Cross-sectional study	Simulated patient method	Medium	7, 9, 14, 18, 20
Puspitasari HP et al. [79]	2011	Indonesia	Cross-sectional study	Simulated patient method	High	7, 14
Hadi U et al. [80]	2010	Indonesia	Cross-sectional study	Simulated patient method	High	7, 14
Llor C et al. [81]	2010	Spain	Prospective study	Simulated patient method	Medium	3, 7, 14, 20
Plachouras D et al. [82]	2010	Greece	Quantitative study	Simulated patient method	Medium	6, 7, 14, 18, 20
Saengcharoen W. et al. [22]	2010	Thailand	Cross-sectional study	Simulated patient method	High	7, 14
Llor C et al. [83]	2009	Spain	Prospective study	Simulated patient method	Medium	3, 7, 14, 20
Rauber C. et al. [84]	2009	Brazil	Cross-sectional study	Pharmacy interviews/questionnaires	High	7, 9, 14
Viberg N et al. [85]	2009	Tanzania	Cross-sectional study	Simulated patient method	High	7, 14, 18
Nyazema N et al. [86]	2007	Zimbabwean	Cross-sectional study	Simulated patient method and Pharmacy interviews/questionnaires	High	7, 13, 14
Caamaño F et al. [87]	2005	Spain	Cross-sectional study	Pharmacy interviews/questionnaires	High	7, 14, 20
Volpato DE et al. [88]	2005	Brazil	Cross-sectional study	Simulated patient method	Medium	7, 9, 10, 20
Caamano Isorna F. et al. [89]	2004	Spain	Cross-sectional study	Pharmacy interviews/questionnaires	High	7, 20
Larson E et al. [90]	2004	United States of America	Cross-sectional study	Simulated patient method	Medium	7, 14, 17, 18, 20
Chalker J et al. [91]	2002	Vietnam	Intervention study	Pharmacy interviews/questionnaires	High	7
Al-Ghamdi MS. [92]	2001	Saudi Arabia	Cross-sectional study	Simulated patient method	Medium	3, 7, 10, 14, 18, 20
Chalker J et al. [16]	2000	Vietnam	Cross-sectional study	Simulated patient method	High	7, 10, 14
Wachter DA et al. [93]	1999	Nepal	Cross-sectional study	Simulated patient method	High	7, 14, 20
Wolffers I. [94]	1987	Sri Lanka	Cross-sectional study	Simulated patient method	Low	3, 5, 7, 9, 10, 11, 12, 14, 18, 20

^a^ Quality Assessment, Quality assessment carried out using the Appraisal tool for Cross-Sectional Studies (AXIS).

#### 2.4.5. Questions and Advice Made at the Time of Dispensing Antibiotics without a Prescription

The information regarding the type of questions and advice given when dispensing antibiotics without a prescription and the resulting percentage for each is summarized in Table 4.

Among the 52 articles that used the simulated patient method, 16 mentioned the question about possible drug allergies. Of these, seven articles presented that none of the pharmacies/pharmacists asked this question at the time of dismissal. The remaining nine had percentages between 8.1% and 59.4%.

Seven articles mentioned the question about the concomitant use of other drugs/medication history and, of these, two papers mentioned that none of the pharmacies/pharmacists did it at the time of dispensation. The remaining articles mentioned percentages ranging from 2.0% to 25.0%.

The article by Hadi U et al. [80] stated that patients were never questioned before dispensing antibiotics without prescription.

Eight articles mentioned the information provided by pharmacies/pharmacists about possible adverse effects, with percentages ranging from 0.0% to 27.9%. Five articles mentioned that none of the pharmacies/pharmacists provided this information.

The information given when dispensing antibiotics without a prescription about drug–drug interactions is mentioned in three of the included articles. Of these, two articles presented percentages of 0.0%. The third article, by Al-Tannir M. et al. [17], showed a growth in the percentage related to this advice, increasing from 0.0% in 2011 to 51.2% in 2018.

Most of the information provided when dispensing without a prescription corresponded to explanations of how to use antibiotics and the duration of treatment.

### 2.5. Study Outcomes—Articles that Used the Pharmacy Interviews/Questionnaires Method

Table 3 synthesizes the extent of antibiotic dispensation without a prescription and additional outcomes of the studies that used the pharmacy interviews/questionnaires method.

#### 2.5.1. Frequency of Antibiotic Dispensation without a Prescription

As in the case of articles using the simulated patient method, the articles where the pharmacy method interviews/questionnaires were applied presented the frequency of dispensation without a prescription in relation to the (i) total number of pharmacies (6 articles); (ii) total number of pharmacists/pharmacy staff (23 articles), (iii) total number of interactions (2 articles) and (iv) number of drugs dispensed without prescription (2 articles).

Of the articles indicating the frequency of dispensing antibiotics without a prescription to the total number of pharmacists, the percentage ranged from 9.1% to 100.0% of pharmacists/pharmacy staff who recognized dispensing antibiotics without a prescription. Four studies presented a percentage of 100.0%, all carried out on Asian countries. Among the articles that presented the percentage about the total number of pharmacists, eight presented the percentage referring to pharmacists who acknowledged dispensing antibiotics without a prescription sometimes/occasionally, with percentages varying from 5.3% to 67.3%.

Of the articles that presented the percentage in relation to the total number of visited pharmacies, the percentage varied from 51.0% in a study conducted in Vietnam in 2002 to 100.0% in two studies, one conducted in Zambia in 2016 and the other conducted in India in 2015.

Among the two articles that presented the percentage in relation to the total number of interactions, it was possible to verify that the percentage varied from 26.4% to 36.4% regarding the interactions that resulted in antibiotic dispensing without a prescription.

The remaining two articles showed the percentage of antibiotics dispensed without prescription in relation to the number of drugs dispensed without a prescription. The article by Paes M.R. et al. [45] showed that in 63.4% of the total dispensing encounters, 5.8% consisted of antibiotics. The article by Nawab A. et al. [60] revealed that out of 100 drugs dispensed without a prescription, 12.2% corresponded to antibiotics.

#### 2.5.2. Name/Class of Antibiotics Most Often Dispensed without a Prescription

The antibiotics most often dispensed without a prescription included amoxicillin (13 articles), cotrimoxazole (3 articles), and amoxicillin-clavulanic acid (3 articles). Some of the articles mentioned the most dispensed antibiotic class, instead of the most dispensed antibiotics, with classes most commonly cited including cephalosporins (3 articles), tetracyclines (2 articles), penicillins (2 articles), and macrolides (2 articles).

#### 2.5.3. Types of Disease/Symptoms Most Commonly Associated with Dispensation without a Prescription

The diseases/symptoms most commonly associated with dispensing antibiotics without a prescription included UTI (9 articles), sore throat (8 articles), cough (7 articles), cold and flu (5 articles), fever (5 articles), diarrhea (4 articles), wound infections (4 articles), and toothache (4 articles).

#### 2.5.4. Questions and Advice Made at the Time of Dispensing Antibiotics without a Prescription

As in the case of articles using the simulated patient method, Table 4 also summarizes the questions and advices provided when dispensing antibiotics without a prescription for articles using the pharmacy interviews/questionnaires method.

The most frequently mentioned question was about the indication of the requested antibiotic (3 articles). The percentages obtained related to this question varied from 36.0% to 94.0%.

Only one article mentioned the question about possible drug allergies, reporting that none of the pharmacists admitted to questioning it before dispensing.

Two articles mentioned information related to possible adverse drug reactions, reporting that the percentage of pharmacists providing this information was of 30.1% [58] and 58.0% [68]. Regarding the articles that mentioned the advice given when dispensing, only three indicated the percentage of pharmacists/pharmacies that do not give any advice or information. This percentage varied from 4.1% to 66.0%.

### 2.6. Study Outcomes—Comparison of the Results Obtained by Using the Two Different Methods

By comparing the results obtained through the two methodologies used, it was possible to verify that the pharmacy interview/questionnaire method showed a higher proportion of articles presenting percentages of antibiotic dispensation without a prescription above 60.0%, when compared to the simulated patient method (pharmacy interview/questionnaire method: 22 of 36 articles; simulated patient method: 29 of 52 articles).

Only 1 of the 52 articles using the simulated patient method recorded that in all pharmacies visited the antibiotic was obtained without a prescription, that is, a percentage of 100.0%. In contrast, six articles using the pharmacy interview/questionnaire method reported that 100.0% of the interviewed pharmacists/pharmacy staff acknowledged dispensing antibiotics without a prescription.

Regarding the antibiotics most commonly dispensed without a prescription, it was possible to verify that amoxicillin was the one most distributed in the two different methods. Moreover, the same also happened with amoxicillin-clavulanic acid. The remaining most dispensed antibiotics differed in the two methods, with articles using a simulated patient method mentioning azithromycin and ciprofloxacin and articles using the pharmacy interview/questionnaire method mentioning cotrimoxazole.

Additionally, it turns out that the diseases/symptoms most used in the simulated patient method were equivalent to the diseases/symptoms most commonly mentioned by pharmacists/pharmacy staff surveyed in the pharmacy interview/questionnaire method.

Table 5 shows the comparison of the frequency of dispensing antibiotics without a prescription obtained within each country using the different methods. The table summarizes the number of studies carried out in the indicated countries, as well as their nature (if a simulated patient method or a pharmacy interview/questionnaire method was used), together with the frequency values of antibiotic dispensing recorded in each study, thus allowing us to analyze the trend of data over time.

For example, it was possible to verify that six articles were conducted in Saudi Arabia, with three using the simulated patient method, 2 using the pharmacy interview/questionnaire method and 1 using both methods. The articles that used the simulated patient method revealed an increase in the frequency of dispensing antibiotics without a prescription from 82.0% [92] to 97.9% [78] and then a slight decrease to 92.2% [34]. The study by Al-Tannir M. et al. [17] registered the lowest value both in 2011 (77.6%) and 2018 (12.5%).

Regarding the articles that used the pharmacy interview/questionnaire method, it was possible to verify that the percentage registered remained at the value of 100.0% for pharmacists who recognized dispensing antibiotics without a prescription [41,68]. Conversely, there was a decrease to 70.7% in the study by Alrasheedy AA. et al. [34]

In Ethiopia, it was possible to observe a decrease in the frequency of dispensing antibiotics without a prescription between studies published in 2016 and 2019, when using the pharmacy interview/questionnaire method [11,56]. However, the same did not happen with studies using the simulated patient method, where it was possible to observe an increase between the study published in 2016 and the studies published in 2019 [11,14,35,37].

When analyzing the data referring to the three articles [11,34,86] that used both methods, it was possible to verify that in two of them [11,34], the highest percentage of antibiotic dispensation without a prescription was associated with the simulated patient method, in the article by Alrasheedy AA. et al. [34] carried out in Saudi Arabia (92.2% versus 70.7%) and the article by Mengistu G et al. [11] conducted in Ethiopia (86.7% versus 50.5%). Alternatively, the article by Nyazema N et al. [86] carried out in Zimbabwe presented a higher percentage of antibiotic dispensation without a prescription associated with the pharmacy interview/questionnaire method (8.0% in the simulated patients method versus 31.0% in the pharmacy interview/questionnaire method).

**Table 2 antibiotics-09-00786-t002:** Study Outcomes—Articles that use the simulated patient method.

Authors (Year)	Sample Size	Frequency of Antibiotic Dispensation without a Prescription	Name/Class of Antibiotics Most Often Dispensed without a Prescription	Types of Disease/Symptoms Most Commonly Associated with Dispensation without a Prescription	Level of Insistence by Patients and the Percentage of Antibiotic Dispensation without a Prescription at Each Level
Al-Tannir M. et al. (2020) [17]	**2011**: 327 PH	**2011**: 77.6%, n = 254	**Sore throat**: Amoxicillin/clavulanic acid (n = 32), azithromycin (n = 11) and amoxicillin (n = 9) **Acute sinusitis**: Amoxicillin/clavulanic acid (n = 22), azithromycin (n = 10), amoxicillin (n = 3), cefaclor (n = 1), ofloxacillin (n = 1) and others (n = 1) **Otitis media**: Amoxicillin/clavulanic acid (n = 12), amoxicillin (n = 8), cephalexin (n = 4), azithromycin (n = 1), cefaclor (n = 1) and cefixime (n = 1) **Acute bronchitis**: Amoxicillin/clavulanic acid (n = 15), amoxicillin (n = 9), azithromycin (n = 5) and others (n = 6) **Diarrhea**: Metronidazole (n = 51), ciprofloxacin (n = 3), cotrimoxazole (n = 1), ofloxacin (n = 1) and others (n = 1) **UTI**: Ciprofloxacin (n = 39), amoxicillin (n = 2), cefixime (n = 1), clarithromycin (n = 1) and others (n = 1)	Sore throat, acute sinusitis, otitis media, acute bronchitis, diarrhea, and UTI	Three levels of demand: level 1 (Can I have something to relieve my symptoms?), level 2 (Can I have something stronger?), level 3 (I would like to have an antibiotic.)
	**2018**: 327 PH	**2018**: 12.5%, n = 41	**Sore throat**: Amoxicillin/clavulanic acid (n = 5), azithromycin (n = 1) and amoxicillin (n = 2) **Acute sinusitis**: Amoxicillin (n = 1) **Otitis media**: Amoxicillin/clavulanic acid **Acute bronchitis**: Amoxicillin/clavulanic acid (n = 2), azithromycin (n = 1), amoxicillin (n = 1) **Diarrhea**: Metronidazole (n = 17) **UTI**: Ciprofloxacin (n = 4), nitrofurantoin (n = 1), sulfamethoxazole (n = 1) and trimethoprim (n = 1)	Sore throat, acute sinusitis, otitis media, acute bronchitis, diarrhea, and UTI	Three levels of demand: level 1 (Can I have something to relieve my symptoms?), level 2 (Can I have something stronger?), level 3 (I would like to have an antibiotic.). All the obtained antibiotics were dispensed under level 3
Bahta M. et al. (2020) [27]	153 V	87.6% **Uncomplicated UTI**: 89.2% **Acute watery diarrhea**: 86.1%	Ciprofloxacin (n = 65, 47.8%), cotrimoxazole (n = 51, 37.5%), amoxicillin (n = 11, 8.1%), doxycycline (n = 5, 3.7%), tinidazole (n = 3, 2.2%) and metronidazole (n = 1, 0.7%). **Uncomplicated UTI**: Ciprofloxacin (n = 38, 56.7%), cotrimoxazole (n = 14, 20.9%), amoxicillin (n = 11, 16.4%), doxycycline (n = 3, 4.5%) and tinidazole (n = 1, 1.5%) **Acute watery diarrhea**: Ciprofloxacin (n = 27, 39.1%), cotrimoxazole (n = 37, 53.6%), doxycycline (n = 2, 2.9%), tinidazole (n = 2, 2.9%) and metronidazole (n = 1, 1.4%)	UTI and acute watery diarrhea	Three levels of demand: level 1 (Asked for some drugs to alleviate the symptoms) (81.3%), level 2 (Request for unspecified antibiotics) (11.2%), level 3 (Ask pharmacy attendant for a specific type of antibiotics) (6.7%)
Chen J. et al. (2020) [28]	1106 PH	83.6%, n = 925	Penicillins (n = 333, 36.0%), cephalosporins (n = 274, 29.6%), macrolides (n = 250, 27.0%)	Mild upper respiratory tract symptoms (young adult)	Three levels of demand: level 1 (Symptoms only described) (25.2%), level 2 (Asked for antibiotics) (52.1%), level 3 (Asked for penicillin or cephalosporins) (6.3%).
Halboup A. et al. (2020) [30]	1000 PH, 200 each scenario	73.3%, n = 733 **Sore throat**: 99.5%, n = 199 **Cough**: 92%, n = 184 **Diarrhea**: 75.5%, n = 151 **Otitis media**: 52%, n = 103 **UTI**: 48%, n = 96	Penicillin (48.3%), sulfonamide (12.5%), macrolide (10.6%), fluoroquinolones (8.8%), chloramphenicol (0.3%)	Sore throat, otitis media, cough, diarrhea, and UTI	Three different levels of demand: level 1 (Asked for medications to relieve the symptoms), level 2 (Asked for a stronger medication), level 3 (Asked for an antibiotic) **Sore throat**: level 1 (n = 195, 98.0%), level 2 (n = 2, 1.0%), level 3 (n = 2, 1.0%) **Cough**: level 1 (n = 14, 7.6%), level 2 (n = 101, 55.0%), level 3 (n = 69, 37.5%) **Diarrhea**: level 1 (n = 121, 80%), level 2 (n = 22, 14.5%), level 3 (n = 8, 5.3%) **Otitis media**: level 1 (n = 78, 75.0%), level 2 (n = 25, 24%), level 3 (n = 1, 1%) **UTI** Level 1 (n = 75, 78%), level 2 (n = 19, 19.8%), level 3 (n = 2, 2.0%).
Shi L. et al. (2020) [31]	147 PH (Pediatric case: 73; Adult case: 74)	88.4%, n = 130 **Pediatric case**: 79.5%, n = 58 **Adult case**: 97.3%, n = 72	**Pediatric case**: Cephalosporin (35.8%), azithromycin (29.6%) and roxithromycin (16.1%) **Adult case**: Azithromycin (30.0%), cephalosporin (28.0%) and roxithromycin (26.0%)	Pediatric and adult acute cough associated with a common cold	Three levels of demand: level 1 (Client required some medicine for cough) (22.5%), level 2 (Client explicitly expressed the requirement of antibiotics) (60.5%), level 3 (Client specifically required roxithromyci) (5.4%) **Pediatric case**: level 1 (n = 15, 20.6%), level 2 (n = 39, 53.4%), level 3 (n = 4, 5.5%). **Adult case**: level 1 (n = 18, 24.3%), level 2 (n = 50, 67.6%), level 3 (n = 4, 5.4%).
Wang X. et al. (2020) [32]	120 PH/V	73.3%, n = 88	Norfloxacin (n = 60), gentamicin (n = 13), levofloxacin (n = 8), ciprofloxacin hydrochloride (n = 8), cefotaxime (n = 1), oxytetracycline (n = 1) and trimethoprim (n = 1)	Acute diarrhea	Two levels of demand: level 1 (“Hi, I have suffered from diarrhea since yesterday, please give me some medicine.”) (55%), level 2 (“Hi, I have suffered from diarrhea since yesterday, and I am here to buy antibiotics.”) (91.7%)
Abdelaziz AI et al. (2019) [33]	238 PH (acute bronchitis: 125, common cold: 113)	98.4% **Acute bronchitis**: 97.6%, n = 122 **Common cold**: 99.1%, n = 112	Amoxicillin: acute bronchitis: 97.6%, common cold: 99.1%.	Acute bronchitis and common cold	
Alrasheedy AA. et al. (2019) [34]	116 PH, 58 each scenario	92.15% **Pharyngitis**: 96.6%, n = 56 **UTI**: 87.7%, n = 50		Pharyngitis and UTI	Three levels of demand: level 1 (Asked for something to relieve the symptoms), level 2 (Asked for something stronger), level 3 (Simulated patient directly requested an antibiotic) **Pharyngitis**: level 1 (85.7%), level 2 (5.4%), level 3 (8.9%) **UTI**: level 1 (74.0%), level 2 (8.0%), level 3 (18.0%)
Chang J. et al. (2019) [7]	2411 PH, 4822 INT	59.3% **Diarrhoea**: 48.5%, n = 1169 **URTI**: 70.1%, n = 1690	Amoxicillin and cephalosporins	Paediatric diarrhoea and adult acute URTI	Three levels of demand: level 1 (“Can you give me some medicine to alleviate the patient’s symptoms?”), level 2 (“Can you give me some antibiotics?”), level 3 (“I would like some amoxicillin or cephalosporins.”) **Diarrhoea**: level 1 (n = 142, 5.9%), level 2 (n = 685, 28.4%), level 3 (n = 342, 14.2%) **URTI**: level 1 (n = 321, 13.3%), level 2 (n = 888, 36.8%) and level 3 (n = 481, 20.0%)
Damisie G et al. (2019) [35]	18 DS	94.4%, n = 17 **Sore throat**: 77.8%, n = 14 **Acute diarrhea**: 88.9%, n = 16 **UTI**: 94.4%, n = 17	**Sore throat**: Amoxicillin (n = 10, 71.4%), azithromycin (n = 3, 21.4%), amoxicillin/azithromycin (n = 1, 7.2%) **Acute diarrhea**: Metronidazole (n = 8, 50.0%), rifampicin (n = 4, 25.0%), tinidazole (n = 2, 12.5%) and ciprofloxacillin/tinidazole/metronidazole (n = 2, 12.5%) **UTI**: Ciprofloxacin (38.90%), norfloxacin (33.30%), cotrimoxazole (16.70%) and amoxicillin (5.60%)	Sore throat, acute diarrhea, and UTI	Three levels of demand: level 1 (Asking something to alleviate his/her symptoms), level 2 (Asking for a stronger medication), level 3 (Clear request for an antibiotic in the case of not achieving the previous two levels of demand) **Sore throat**: level 1 (n = 14, 100%), level 2, level 3 (0%) **Acute diarrhea**: level 1 (n = 11, 68.75%), level 2 (n = 3, 18.75%), level 3 (n = 2, 12.5%) **UTI**: level 1 (n = 16, 94.1%), level 2 (n = 1, 5.9%), level 3 (0%)
Koji EM et al. (2019) [37]	262 PH	63.4%, n = 166	Amoxicillin, amoxicillin-clavulanate, azithromycin, trimethoprim/sulfamethoxazole, metronidazole, ceftriaxone, cloxacillin, vancomycin, ampicillin, cefotaxime, gentamicin	Common cold, acute onset diarrhea, pneumonia (child), meningitis (child)	
Mengistu G et al. (2019) [11] ^a^	105 PH	86.7%	Cotrimoxazole (97.8%)	Acute watery diarrhea (child)	
Nafade V. et al. (2019) [19]	279 PH, 1522 INT (761 each scenario)	4.0% **Adult cases**: 4.3%, n = 33 **Child cases**: 2.9%, n = 22	**Children** **Fever**: Bromhexine (n = 1), cetirizine (n = 1), chlorpheniramine (n = 7), levocetirizine (n = 1), montelukast (n = 1) URTI: Bromhexine (n = 8), cetirizine (n = 11), chlorpheniramine (n = 45), dextromethorphan (n = 45), levocetirizine (n = 2), levosalbutamol (n = 2), montelukast (n = 1), salbutamol (n = 2), terbutaline (n = 1) **Diarrhoea**: Loperamide (n = 23) **Adults****Fever**: Bromhexine (n = 9), cetirizine (n = 65), chlorpheniramine (n = 53), dextromethorphan (n = 4), levocetirizine (n = 3), nimesulide (n = 23) **URTI**: Bromhexine (n = 109), cetirizine (n = 8), chlorpheniramine (n = 115), dextromethorphan (n = 62), levocetirizine (n = 13), montelukast (n = 5), salbutamol (n = 1), terbutaline (n = 26) **Diarrhoea**: Domperidone (n = 2), loperamide (n = 227), omeprazole (n = 3), ranitidine (n = 2)	URTI, uncomplicated acute diarrhoea, and acute febrile illness suggestive of malaria	
Zawahir S et al. (2019) [38]	242 PH	41.0%, n = 99 **Sore throat**: 43.0% **Common cold**: 15.0% **Diarrhoea**: 50.0% **UTI**: 55.0%	**Sore throat**: Erythromycin (n = 17, 65%), azithromycin (n = 7, 27%;), ciprofloxacin (n = 1, 4%), amoxicillin (n = 1, 4%) **Common cold**: Amoxicillin (89%; n = 8) and others (n = 1, 11%) **Diarrhoea**: Metronidazole (n = 23, 79%), ciprofloxacin (n = 2, 7%), erythromycin (n = 2, 7%), azithromycin (n = 1, 3%) and others (n = 1, 3%). **UTI**: Ciprofloxacin (n = 26, 76%), norfloxacin (n = 5, 15%), others (n = 3, 9%)	Viral infections: acute sore throat, common cold (child), and acute diarrhoea and bacterial uncomplicated UTI	Three levels of demand: level 1 (Can I get some medicine to alleviate the symptoms?) (n = 39, 16%), level 2 (Can I get something stronger?) (n = 33, 14%), level 3 (I would like an antibiotic.) (n = 27, 11%). **Sore throat**: level 1 (n = 11, 18%), level 2 (n = 7, 12%), level 3 (n = 8, 13%) **Common cold**: level 1 (n = 1, 2%), level 2 (n = 6, 10%), level 3 (n = 2, 3%) **Diarrhoea**: level 1 (n = 9, 15%), level 2 (n = 11, 18%), level 3 (n = 10, 17%) **UTI**: level 1 (n = 18, 29%), level 2 (n = 9, 15%), level 3 (n = 7, 11%)
Erku D.A. et al. (2018) [43]	50 CMRO, 100 V	86.0%, n = 86 **Childhood diarrhea**: n = 40 (Pharmacy: (75%, n = 21/28); drug store: 86.4%, n = 19/22); **URTI**: n = 46 (Pharmacy: 85.7%, n = 24/28; drug store: 100%, n = 22/22)	**Childhood diarrhea**: Cotrimoxazole (n = 11), metronidazole (n = 15) **URTI**: Amoxicillin (n = 23), amoxicillin-clavulanic acid (n = 19), azithromycin (n = 15), ciprofloxacin (n = 5), cephalexin (n = 1), cefexime (n = 1), levofloxacin (n = 3)	Acute childhood diarrhea and URTI	
Horumpende PG et al. (2018) [20] ^b^	82 PH (26 class I, 56 class II)	92.3% **Fever**: 100.0% **Diarrhoea**: 100.0% **Runny nose**: 100.0% **Painful urination**: 88.8% **Cough**: 75.0% **Pharmacies class I**: 44.4%; **Pharmacies class II**: 56.0%;	Amoxycillin (n = 1), ampiclox (n = 3), trimethoprim/sulphamethoxazole (n = 2), cefixime (n = 1), amoxyclav, (n = 1), azithromycin (n = 4), erythromycin (n = 1), metronidazole	Fever, diarrhoea, runny nose, painful urination, and cough	
Ibrahim IR et al. (2018) [12] ^a^	75 PH	20.0%, n = 15	Metronidazole (n = 4, 5.3%), furazolidone (n = 3, 4.0%)	Acute diarrhea	
Mohamed Ibrahim M.I. et al. (2018) [44]	25 PH (common cold: 15 PH, 30 V; allergic rhinitis: 10 PH, 20 V)	5.0% **Common cold**: 5.0% and **Allergic rhinitis**: 5.0%		Common cold (signs and symptoms: sore throat, slight cough, tiredness and body aches), and allergic rhinitis (signs and symptoms: running nose or congestion, sneezing, slight sore throat with phlegm and slight cough when in bed)	
Zapata-Cachafeiro M et al. (2018) [48]	977 PH	18.8%, n = 184	Amoxicillin (n = 127, 69.0%), amoxicillin-clavulanic acid (7%), azithromycin (n = 42, 22.8%), cotrimoxazole (n = 7, 3.8%), moxifloxacin (n = 4, 2.2%), cefuroxime (n = 2, 1.1%), clarithromycin (n = 1, 0.5%) and clindamycin (n = 1, 0.5%)	Sore throat, difficulty swallowing, and feeling feverish, in addition to congestion and cough	Four levels of demand: level 1 (Request for medication to relieve the symptoms) (2.97%), level 2 (Request for a stronger medication than that offered) (4.22%), level 3 (Request for an antibiotics) (4.52%), level 4 (Specific request for amoxicillin) (8.54%)
Zawahir S et al. (2018) [49]	242 PH	61.0%, n = 147	Ciprofloxacin (n = 44, 70%), erythromycin, metronidazole, amoxicillin (n = 32, 52%)	URTI (adult and child), watery diarrhoea and UTI	
Chang J. et al. (2017) [52]	256 PH	66.8% **Paediatric diarrhoea**: 55.9%, n = 143 **Adult acute URTI**: 77.7%, n = 199		Paediatric diarrhoea and adult acute URTI	Three levels of demand: level 1 (“Can you give me some drugs to alleviate the symptoms of the disease?”), level 2 (“Can you give me some antibiotics?”), level 3 (“I would like some amoxicillin or cefaclor.”) **Paediatric diarrhoea**: Level 1 (9%), level 2 (37.5%), level 3 (9.4%) **Acute URTI**: level 1 (26.2%), level 2 (43.0%), level 3 (8.6%)
Jaisue S. et al. (2017) [21] ^b^	91 class I PH	68.1%, n = 62	Furazolidone (n = 31, 34.1%), nifuroxazide (n = 17, 18.7%), cotrimoxazole (n = 10, 11.0%), metronidazole (n = 2, 2.2%), cephalexin (n = 2, 2.2%), azithromycin (n = 1, 1.1%), norfloxacin (n = 1, 1.1%)	Non-infectious diarrhoea in a 14-month-old child	
Marković-Peković V et al. (2017) [18]	**2010**: 131 PH	**2010**: 58.0%, n = 76	**2010**: Amoxicillin (n = 65, 85.5%), ampicillin (n = 5, 6.6%), cephalexin (n = 2, 2.6%) and doxycycline (n = 4, 5.3%)	URTI	
	**2015**: 383 PH	**2015**: 18.5%, n = 71	**2015**: Amoxicillin (n = 57, 80.3%), ampicillin (n = 9, 12.7%), cephalexin (n = 3, 4.2%), azithromycin (n = 1, 1.4%) and amoxicillin and enzyme inhibitor (n = 1, 1.4%)	URTI	
Okuyan B. et al. (2017) [54]	70 PH	Antibiotic + NSAIDs (n = 15, 21.4%) Antibiotic alone (n = 17, 24.2%)	Cefuroxime alone (n = 5), cefuroxime with other medication (n = 6), amoxicillin-clavulanic acid alone (n = 12), amoxicillin-clavulanic acid with other medication (n = 9)	Acute uncomplicated rhinosinusitis	
Abegaz T.M. et al. (2016) [14] ^a^	113 PH	51.3%, n = 58	Cotrimoxazole and metronidazole	Acute diarrhea	
Guinovart MC et al. (2016) [57]	220 PH	54.1%, n = 119	β-lactam antibiotic and amoxicillin-clavulanic acid	UTI, sore throat, and acute bronchitis	Four levels of demand: level 1 (Medication to treat the symptoms was required), level 2 (A stronger medication was required), level 3 (An antibiotic was required), level 4 (A specific antibiotic was required: amoxicillin/clavulanic acid for a UTI and amoxicillin for a sore throat and acute bronchitis)
Ibrahim M.I.B.M. et al. (2016) [13] ^a^	30 PH 60 INT, 95 DD	43.2%, n = 41	Nifuroxazide (n = 21, 22.11%), metronidazole alone (n = 12, 12.63%), metronidazole with other medication (n = 4, 4.21%), tinidazole (n = 2, 2.11%), furazolidone (n = 1, 1.05%)	Acute gastroenteritis	
Satyanarayana S. et al. (2016) [61]	622 PH, 1200 V	27.0%, n = 319	**Presumed tuberculosis**: Amoxicillin (n = 100), ofloxacin (n = 25), ciprofloxacin (n = 24), azithromycin (n = 23), cefixime (n = 19), levofloxacin (n = 14), ampicillin (n = 8), roxithromycin (n = 8), cloxacillin (n = 6), erythromycin (n = 6) **Tuberculosis case with positive sputum report**: Amoxicillin (n = 50), azithromycin (n = 16), cefixime (n = 11), levofloxacin (n = 7), ofloxacin (n = 7), ciprofloxacin (n = 5)	Tuberculosis	
Almaaytah A et al. (2015) [62]	202 PH (Sore throat: 41, otitis media: 38, acute sinusitis: 39, diarrhea: 42, UTI: 42)	74.3%, n = 150 **Sore throat**: 97.6%, n = 40 **Otitis media**: 68.4%, n = 26 **Acute sinusitis**: 48.5%, n = 15 **Diarrhea**: 81.0%, n = 34 **UTI**: 83.3%, n = 35	**Sore throat**: Penicillins (45%), penicillin/penicillinase inhibitor (32.5%), first-Gen cephalosporins (10%) **Otitis media**: choramphenicol (19.2%), penicillins (15.4%), macrolides (11.5%) **Acute sinusitis**: Penicillins (20%), fluoroquinolones (20%), macrolides (20%) **Diarrhea**: antiprotozoals (79.4%), sulfonamides (5.9%), fluoroquinolones (2.9%) **UTI**: fluoroquinolones (80%), third-Gen cephalosporins (5.7%)	Sore throat, otitis media, acute sinusitis, diarrhea, and UTI	Three levels of demand: level 1 (Asking for something to alleviate the symptoms) (n = 121, 59.9%), level 2 (Asking for a stronger medication) (n = 4, 2%), level 3 (Clear request for an antibiotic) (n = 25, 12.4%) **Sore throat**: level 1 (n = 39, 95.1%), level 2 (n = 0), level 3 (n = 1, 2.4%) **Otitis media**: level 1 (n = 25, 65.8%), level 2 (n = 0), level 3 (n = 1, 2.6%) **Acute sinusitis**: level 1 (n = 3, 7.7%), level 2 (n = 4, 10.3%), level 3 (n = 8, 20.5%) **Diarrhea**: level 1 (n = 24, 57.1%), level 2 (n = 0) and level 3 (n = 10, 23.8%) **UTI**: level 1 (n = 30, 71.4%), level 2 (n = 0), level 3 (n = 5, 11.9%)
Shet A et al. (2015) [65]	261 PH **URTI**: 115 PH **Acute gastroenteritis (child)**: 146 PH	66.7%, n = 174 **URTI**: 71.3%, n = 82; **Acute gastroenteritis (child)**: 63.0%, n = 92	**URTI**: Amoxicillin (n = 42, 51.2 %), amoxicillin-clavulanate (n = 2, 2.4%), ampicillin-cloxacillin (n = 4, 4.9%), cephalexin (n = 2, 2.4%), cefixime (n = 2, 2.4%), azithromycin (n = 10, 12.2%), roxithromycin (n = 3, 3.7%), ciprofloxacin (n = 10, 12.2%), levofloxacin (n = 3, 3.7%) and ofloxacin (alone) (n = 4, 4.9%) **Acute gastroenteritis (child)**: Ofloxacin (alone) (n = 7, 7.6%), norflaxacin (alone) (n = 8, 8.7%), norfloxacin + metronidazole (n = 38, 41.3%), ofloxacin + metronidazole (n = 9, 9.8%), ofloxacin + ornidazole (n = 7, 7.6%), metronidazole (alone) (n = 14, 15.2%) and furazolidone (n = 9, 9.8%)	URTI (adult) and acute gastroenteritis (child)	Two levels of demand: level 1 (Request for a “medicine” to alleviate the described symptoms) (55.6%), level 2 (Specifically asked for a “stronger” medicine) (44.4%) **URTI infection**: level 1 (53.9%), level 2 (17.4%) **Acute gastroenteritis (child)**: level 1 (56.8%), level 2 (6.2%)
Alabid A.H.M.A. et al. (2014) [67]	50 PH/Ph, 100 V	32.0%, n = 32	Amoxil and Amoxiclav (n = 11, 11.0%), erythromycin (n = 9, 9.0%), cefalexin (n = 4, 4.0%)	Common cold symptoms (symptoms of URTI)	
Malik M. et al. (2013) [15] ^a^	238 PH/V	Simulated patients were treated in 198 V (83.1%); antibiotics were given in 69 V (34.4%)		Uncomplicated malaria fever	
Minzi O et al. (2013) [23]	85 ADDO and 60 DLDB	67.0% ADDO: 79.0% DLDB: 55.0%	Ciprofloxacin, amoxicillin, ampicillin, chloramphenicol, procaine penicillin, tetracycline	Cough, headache and diarrhea (“typhoid”, child), injured on the left hip by a piece of metal, fever and diarrhea (“cholera”, child), vomiting and diarrhea (“typhoid”, child), cough (“Pneumonia”), UTI, complaining of yellowish urethral discharge with a bad smell (“gonorrhea”)	
Marković-Peković V et al. (2012) [74]	131 PH	58.0%, 76 pharmacies	Amoxicillin (85%), doxycyline (5%), ampicillin (7%), and cefalexin (3%)	URTI	Without insistence in case of refusal
Rathnakar U.P. et al. (2012) [75]	60 PH, 20 for each scenario	51.7%, n = 31	Amoxicillin (n = 19, 31.7%), erythromycin (n = 1, 1.7%), ampicillin+cloxacillin (n = 1, 1.7%), azithromycin (n = 4, 6.7%)	URTI, acute bronchitis, and diarrhoea	Two levels of demand: Level 1 (Can I have something for my symptoms?) and level 2 (I would like an antimicrobial agents.) **URTI**: level 1 (35%), levels 2 and 3 (40%) **Acute bronchitis**: level 1 (20%), level 2 and level 3 (30%) **Diarrhoea**: level 1 (20%), levels 2 and 3 (10%)
Simó S et al. (2012) [76]	50 PH	8.0%, n = 4	Amoxicillin/clavulanic acid (n = 4, 8%)	URTI symptoms and fever	
Al-Faham Z et al. (2011) [77]	200 PH	97.0%, n = 194	Amoxicillin/clavulanic acid 1000 mg (n = 73, 37.6%); amoxicillin (n = 45, 23.1%); amoxicillin/clavulanic acid 625 mg (n = 25, 12.8%); amoxicillin/floxacillin (n = 13, 6.7%), cefodroxil (n = 13, 6.7%), clarithromycin (n = 6, 3.0%), azithromycin (n = 5, 2.5%), ciproflaxacillin (n = 4, 2.0%), Cloxacillin/Ampicillin (n = 4, 2.0%), cefixime (n = 2, 1.0%), cefprodoxime 100mg (n = 2, 1.0%) and cefprodoxime 200mg (n = 2, 1.0%)	Sinusitis (fever, runny nose with clear secretion and a headache in the frontal sinus region)	Two levels of demand: level 1 (without insistence) (n = 174, 87%), level 2 (with insistence) (n = 20, 10%)
Al-Mohamadi A et al. (2011) [78]	60 PH/Ph	97.9%	Co-amoxiclav (Augmentin), amoxicillin-clavulanic acid, cefaclor	Sore throat	
Puspitasari HP et al. (2011) [79]	264 PH/V, 88 for each scenario	91.0%, n = 80	Ciprofloxacin (n = 80, 91%), tetracycline (n = 80, 91%), amoxicillin (n = 74, 84%)	“discomfort on urination”, infected leg wounds and “productive cough, rainy nose, fever and lost of appetite”	
Hadi U et al. (2010) [80]	104 MRO (75 PH, 28 K, 1 DS)	75.9%, n = 79	Amoxicillin (n = 15), chloramphenicol (n = 18), ciprofloxacin (n = 14), cotrimoxazole (n = 14), tetracycline 250 mg (n = 15), tetracycline 500 mg (n = 2)		
Llor C et al. (2010) [81]	197 PH (sore throat: 69, acute bronchitis: 59, UTI: 69)	45.2%, n = 89 **Sore throat**: 34.8%, n = 24 **Acute bronchitis**: 16.9%, n = 10 **UTI**: 79.7%, n = 55		Sore throat, acute bronchitis, and UTI	Three levels of demand: level 1 (Asked for something to alleviate the symptoms of the infection), level 2 (“This medication is not very strong, can’t you give me something stronger?”), level 3 (Asking openly for an antibiotic)
Plachouras D et al. (2010) [82]	174 PH/V (ciprofloxacin: 102, amoxicillin + clavulanic acid: 72)	72.4%, n = 126 **Ciprofloxacin**: 53.0%, n = 54 **Amoxicillin + clavulanic acid**: 100.0%, n = 72	Ciprofloxacin (n = 54, 53%), amoxicillin/clavulanic acid (n = 72, 100%)		Without insistence in case of refusal
Saengcharoen W. et al. (2010) [22] ^b^	115 PH (type I: 96 and type II: 19)	**Antibiotics + ORS** I (non-Ph + Ph): 26.0%, n = 25 I (non-Ph): 30.3%, n = 10 II: 21.1%, n = 4 **Antibiotics only** I (non-Ph + Ph): 21.9%, n = 21 I (non-Ph): 27.3%, n = 9 II: 26.3%, n = 5 **Antibiotics+ combined drugs** I (non-Ph + Ph): 1.0%, n = 1 I (non-Ph): 3.0%, n = 1 II: n = 1, 5.3%	Pharmacy personnel: Nifuroxazide, cotrimoxazole, norfloxacin, erythromycin and amoxicillin	Acute childhood diarrhoea	
Llor C et al. (2009) [83]	197 PH (sore throat: 69, acute bronchitis: 59, UTI: 69)	45.2%, n = 89 **Sore throat**: 34.8%, n = 24 **Acute bronchitis**: 16.9%, n = 10 **UTI**: 79.7%, n = 55	**Sore throat**: Amoxicillin (n = 21, 87.5%), amoxicillin/clavulanic acid (n = 2, 8.3%), azithromycin (n = 1) **Acute bronchitis**: Amoxicillin (n = 10) **UTI**: Norfloxacin (n = 22, 40.0%), fosfomycin trometamol (n = 20, 36.4%), pipemidic acid (n = 8, 14.5%)	Sore throat, acute bronchitis, and uncomplicated UTI	Three levels of demand: level 1 (“Can you give me something to alleviate the symptoms of the infection?”) (n = 65, 33.0%), level 2 (“Can’t you give me something stronger?”) (n = 17, 8.6%), level 3 (“I would like an antibiotic.”) (n = 7, 3.6%) **Sore throat**: level 1 (n = 12, 17.4%), level 2 (n = 10, 14.5%), level 3 (n = 2, 2.9%) **Acute bronchitis**: level 1 (n = 1, 1.7%), level 2 (n = 5, 8.5%), level 3 (n = 4, 6.8%) **UTI**: level 1 (n = 52, 75.4%), level 2 (n = 2, 2.9%), level 3 (n = 1, 1.4%)
Viberg N et al. (2009) [85]	SCM-female: 144 V and SCM-male: 107 V	55.5% SCM-female: 35.0% SCM-male: 76.0%	**SCM-female**: Doxycycline (n = 23), amoxicillin (n = 5), sulfamethoxazole/trimethoprim (n = 13), erythromycin (n = 1), ciprofloxacin (n = 15), metronidazole (n = 16), nitrofurantoin (n = 1) **SCM-male**: Doxyxycline (n = 40), amoxicillin (n = 3), phenoxymethylpenicillin (n = 2), sulfamethoxazole/trimethoprim (n = 13), erythromycin (n = 1), ciprofloxacin (n = 30), metronidazole (n = 12), nitrofurantoin (n = 1)	Abnormal vaginal discharge and itching (SCM-female), urethral discharge (SCM-male)	
Nyazema N et al. (2007) [86]	STI female: 57 V STI male: 63 V Acute diarrhoea: 68 V	8.0% STI female: 7.0% STI male: 8.0% Acute diarrhoea: 9.0%		Vaginal discharge and itching (STI female), urethral discharge (STI male) and acute diarrhoea (child)	
Volpato DE et al. (2005) [88]	107 PH	74.0%	Amoxicillin (n = 46, 74%), azythromycin (n = 6, 9.6%), sulfamethoxazole/trimethoprim (n = 5, 8.1%), cephalexin (n = 2, 3.2%), erythromycin (n = 2, 3.2%) and ampicillin (n = 1, 1.6%)	Acute and uncomplicated rhino-sinusitis	Three levels of demand: No insistence (58%), insisting once (13%) or twice (3%) when the antibiotic was denied.
Larson E et al. (2004) [90]	101 DS (PHN: 34, PBNHN: 37, PWNHN:30)	50.0% **PHN**:100.0%, n = 34 **PBNHN** and **PWNHN**: 0.0%	Ampicillin (n = 26, 76.5%), ampicillin and tetracycline (n = 2, 5.9%), ampicillin and erythromycin; amoxicillin (n = 2, 5.9%), erythromycin (n = 1, 2.9%)	Sore throat	
Al-Ghamdi MS. (2001) [92]	88 PH	82.0%, n = 72	Fluoroquinolones (First choice: n = 50, 69% and Second choice: n = 59, 87%), cotrimoxazole (First choice: n = 9, 13% and Second choice: n = 3, 4%), penicillins (First choice: n = 8, 11% and Second choice: n = 3, 4%), cephalosporins (First choice: n = 3, 4% and Second choice: n = 1, 2%), tetracyclins (First choice: n = 2, 3% and Second choice: n = 2, 3%)	Uncomplicated lower, UTI	
Chalker J et al. (2000) [16] ^a^	60 PH, 297 V	81.5%, n = 242	Tetracyclines (n = 36), amphenicols (n = 14), β-lactam antibacterials- Penicillins (n = 10), other β- lactam antibacterials (16), sulphonamides/trimethoprim (n = 6), macrolides and lincosamides (n = 15), quinolones (n = 188), metronidazole (n = 3), spectinomycin (n = 4), drugs for treatment of TB (n = 1)	STD	
Wachter DA et al. (1999) [93]	100 PH	67.5% Dysuria: 38% Diarrhoea: 97%	**Dysuria**: Norfloxacin (28%), amoxicillin (5%), trimethaprim/sulphamethoxazole (2%), nalidixic acid (2%), ciprofloxacin (1%) **Diarrhoea**: Metronidazole (62%), metronidazole/diloxanide furoate (24%), metronidazole/nalidixic acid (comb) (9%), metronidazole/nalidixic acid (separate) (2%)	Dysuria and acute watery diarrhoea (child)	
Wolffers I. (1987) [94]	28 PH	100.0%	Tetracyclin (100%)		

Sample size: PH, Pharmacies; Ph, Pharmacists; V, Visits; DS, DrugStores; MRO, Medicine Retail Outlets; INT, Interactions; DD, Drugs Dispensed; ADDO, Accredited Drug Dispensing Outlets; DLDB, Duka la Dowa Baridi; K, Kiosk; PHN, Primarily Hispanic Neighborhood; PBNHN, Primarily Black Non- Hispanic Neighborhood; PWNHN, Primarily White Non-Hispanic neighborhood; Types of disease/symptoms most commonly associated with dispensation without a prescription: UTI, Urinary Tract Infection; URTI, Upper Respiratory Tract Infection; STD, Sexually Transmitted Diseases; ^a^ Articles that do not explicitly mention throughout the full article that it is about dispensing antibiotics without a prescription, but rather the management of diseases/symptoms in pharmacies/drug stores; ^b^ Articles showing the frequency of dispensing antibiotics without a prescription for class I and class II pharmacies. Class I pharmacies are legally authorized to dispense antibiotics without a prescription.

**Table 3 antibiotics-09-00786-t003:** Study Outcomes—articles that used the Pharmacy interviews/questionnaires.

Authors (Year)	Sample Size	Frequency of Antibiotic Dispensation without a Prescription	Name/Class of Antibiotics Most Often Dispensed without a Prescription	Types of Disease/Symptoms Most Commonly Associated with Dispensation without a Prescription
Abubakar U. et al. (2020) [24]	98 Ph	PD: 74.5% (n = 73) PSD: 67.3% (n = 66)	Penicillin (n = 84, 85.7%), tetracycline (n = 69, 70.4%), cephalosporin (n = 63, 64.3%), quinolone (n = 61, 62.2%), macrolides (n = 54, 55.1%), sulphonamides (n = 47, 48.0%), aminoglycosides (n = 34, 34.7%), carbapenems (n = 13, 13.3%.	UTI, typhoid fever, genital infections (gonorrhea), wound infections, eye infections, ear infections, diarrhea, malaria, toothache, and cold/flu
Abubakar U. (2020) [25]	98 Ph	PD: 96.9% (n = 95) PSD: 60.2% (n = 59)		
Badro D.A. et al. (2020) [26]	250 Ph	88.0% acknowledged dispensing medications without a prescription, those medications included antibiotics (60.0%)		
Gajdács M. et al. (2020) [29]	192 Ph	PD: 26.0%		
Alrasheedy AA. et al. (2019) [34]	116 PH	70.7%, n = 82		
Hallit S et al. (2019) [36]	280 PH, 202 Ph	84.6%		Pharyngitis, otitis media, diarrhoea, and vomiting (child)
Mengistu G et al. (2019) [11] ^a^	105 PH/PS	50.5%		Acute watery diarrhea (child)
Zawahir S. et al. (2019) [39]	265 PS	31.7%, n = 84 Non-Ph: 33.3%, n = 18; Ph: 31.6%, n = 66		Acute sore throat, common cold, acute diarrhoea, wound infection, or uncomplicated UTI
Ajie A.A.D. et al. (2018) [40]	190 PH	PD: 92.1% (n = 175)	Amoxicillin (n = 175, 92.1%), cotrimoxazole (n = 175, 92.1% and ciprofloxacin (n = 159, 83.7%)	
Alhomoud F et al. (2018) [41]	20 Ph	100.0%	Amoxicillin/clavulanic Acid (Augmentin), amoxicillin and azithromycin.	Fever, sore throat, cold/flu, and cough.
Awosan KJ et al. (2018) [42]	197 PS	PD: 91.9% (n = 181) PSD: 10.2% (n = 20)		
Paes M.R. et al. (2018) [45]	101 Ph	Dispensing without a prescription was 63.4% of the total dispensing encounters, those medications included antibiotics (5.8%)		
Rehman IU et al. (2018) [46]	181 Ph	PD: 68.0% (n = 123) PSD: 20.4% (n = 37)		
Sarwar M.R. et al. (2018) [47]	400 Ph	PD: 93.7% (n = 375) PSD: 34.5% (n = 138)		
Ansari M. (2017) [50]	16 PH	66.5%	Cephalosporins, penicillins, and macrolides	Respiratory tract complications (e.g., cough), fever, and UTI
Barker A.K. et al. (2017) [51]	24 PS	100.0%		Colds, viral infections, coughs, and sore throat
Mansour O et al. (2017) [53]	173 PH	85.5%		Tonsillitis
Abood EA et al. (2016) [55]	170 Ph	25.3%, n = 43	Amoxicillin (7.3%)	
Erku D.A. (2016) [56]	389 Ph	PD: 90.2% PSD: 5.4% (n = 21)		
Kalungia AC et al. (2016) [58]	73 PH	100%	Amoxicillin (n = 38, 52.1%), cotrimoxazole (n = 18, 24.7%), metronidazole (n = 17, 23.3%)	
Khan M.U. et al. (2016) [59]	188 Ph	PD: 63.3% (n = 119) PSD: 5.3% (n = 10)		
Nawab A. Et al. (2016) [60]	50 PH	12.2% of 100 drugs dispensed without a prescription	Metronidazole and amoxicillin/clavulanate potassium	
Bahnassi A. (2015) [63]	147 Ph	100.0%	Amoxicillin, amoxicillin/clavulanic acid, cephalexin	Sore throat and UTI
Dorj G. et al. (2015) [64]	61 PS	PSD: 21.7% (n = 13) PD’:35.0%	Aminopenicillins, oral (n = 73, 29.9%); aminopenicillins, injection (n = 44, 24.0%); quinolone, oral (n = 30, 24.6%); quinolone, injection (n = 13, 21.3%); cefalosporin, oral (n = 14, 23.0%); cefalosporin, injection (n = 10, 16.4%); macrolides, oral (n = 53, 29.0%); macrolides, injection (n = 29, 15.8%); tetracycline, oral (n = 19, 15.6%) and sulfonamid, oral (n = 18, 29.5%).	Mild/moderate community-acquired pneumonia
Shreya Svitlana A. et al. (2015) [66]	100 PH	100.0%	Cefodoxime (n = 52), amoxicillin (n = 30), doxycycline-doxy (n = 8), cefixime-taxim (n = 10)	Mild toothache
Bahnassi A. (2014) [68]	54 Ph	100.0%	Amoxicillin, amoxicillin/clavulanic acid, azithromycin	Sore throat, sinusitis (pregnant), UTI, ear infection (child), and skin infection
Farah R et al. (2014) [69]	100 Ph	32.0%		Gastrointestinal symptoms, Genito-urinary symptoms, and Respiratory symptoms
Gastelurrutia M.A. et al. (2014) [70]	152 PH	9.8% of the total number of antibiotics dispensed		
Sabry NA et al. (2014) [71]	36 PH, 1158 INT	36.4% Upon pharmacist’s recommendation: 13.1%, n = 152 Upon patient´s request: 23.3%, n = 270	Upon pharmacist´s recommendation: Amoxicillin/fluoxacillin (n = 10, 6.58%), ampicillin/sulbactam (n = 10, 6.58%), amoxicillin (n = 8, 5.26%), co-amoxiclav (n = 11, 7.24%), cephalexin (n = 21, 13.82%), cephradine (n = 10, 6.58%), cefaclor (n = 7, 4.61%), cefuroxime (n = 5, 3.29%), cefoperazone (n = 8, 5.26%), cefotaxime (n = 7, 4.61%), ceftriaxone (n = 7, 4.61%), oxycycline (n = 7, 4.61%), tetracycline (n = 3, 1.97%), clarithromycin (n = 4, 2.63%), clindamycin (n = 9, 5.92%), co-trimoxazole (n = 5, 3.29%), ciprofluxacin (n = 7, 4.61%), gatifluxacin (n = 8, 5.26%) and moxifloxacin (n = 5, 3.29%). Upon patient´s request: Amoxicillin/fluoxacillin (n = 28, 10.37%), co-amoxiclav (n = 24, 8.89%), amoxycillin (n = 44, 16.30%), azatreonam (n = 4, 1.48%), doxycycline (n = 20, 7.40%), tetracyclin (n = 4, 1.48%), cephadrin (n = 13, 4.81%), cephalexin (n = 3, 1.11%), cefadroxil (n = 12, 4.44%), cefuraxime (n = 2, 0.74%), cefixime (n = 10, 3.70%), cefotaxime (n = 4, 1.48), ceftriaxone (n = 4, 1.48), azithromycin (n = 12, 4.44%), spiramycin (n = 8, 2.96%), roxithromycin (n = 4, 1.48%), erythromycin (n = 8, 2.96%), clindamycin (n = 8, 2.96%), co-trimoxazole (n = 6, 2.22%), ciprofloxacin (n = 6, 2.22%), sparfloxacin (n = 5, 1.85%), moxifloxacin (n = 5, 1.85%), levofloxacin (n = 8, 2.96%), ofloxacin (n = 8, 2.96%), entamycin (n = 4, 1.48%), chloramphenicol (n = 3, 1.11%), fusidic acid (n = 8, 2.96%), neomycin/bacitracin (n = 5, 1.85%).	Upon pharmacist´s recommendation: UTI (n = 25, 17.86%), sore throat (n = 24, 17.10%), cold & flu (n = 16, 11.40%), toothache (n = 13, 9.29%), infected wound (n = 11, 7.86%), rhinitis (n = 8, 5.70%), acne (n = 8, 5.70%), abdominal cramps (n = 7, 5.00%), tonsillitis (n = 5, 3.57%), post nasal discharges (n = 4, 2.90%), burning (n = 4, 2.90%), asthma (n = 4, 2.90%), food poisoning (n = 4, 2.90%), stomachache (n = 4, 2.90%) and fracture (n = 3, 2.14%). Upon patient´s request: Fever (n = 36, 15.52%), sore throat (n = 25, 10.78%), tonsillitis (n = 19, 8.19%), gingivitis (n = 16, 6.9%), UTI (n = 12, 5.17%), ear ache & inflammation (n = 12, 5.17%), acne (n = 9, 3.88%), wound infection (n = 8, 3.45%), toothache (n = 8, 3.45%), headache (n = 7, 3.02%), urticaria (n = 7, 3.02%), vomiting (n = 6, 2.59%), food poison (n = 6, 2.59%), otitis (n = 6, 2.59%), flu (n = 5, 2.16%), vaginites (n = 5, 2.16%), sneezing (n = 4, 1.72%), sinusitis (n = 4, 1.72%), cough (n = 4, 1.72%), difficulty in breathing (n = 4, 1.72%), skin infection (n = 4, 1.72%), dandruff (n = 4, 1.72%), endometritis (n = 4, 1.72%), prostatitis (n = 4, 1.72%), dizziness (n = 4, 1.72%), nones/joint infection (n = 4, 1.72%), abortion (n = 3, 1.29%), and irritation (n = 2, 0.86%).
Zapata-Cachafeiro M. et al. (2014) [72]	286 Ph	64.7%, n = 185		Urinary and dental infections
Abasaeed AE et al. (2013) [73]	20 Ph, 1645 INT	26.4%	Ceftriaxone (53.3%), amoxicillin (47.8%), and co-amoxiclav (33.6%)	Cough, influenza, respiratory tract infections, STD, and Helicobacter pylori
Rauber C. et al. (2009) [84]	46 Ph	85.0%		Diseases: Throat infection, UTI, ear infection, sinusitis, pharyngitis, upper airway infection, pneumonia, fever, dental infection, throat plaque, clear and simple infection, skin infection, intestinal infection, cough with secretion, oral infection, and acne. Symptoms: High fever, formation of throat plaques, pus, pain, sore throat, sinusitis, headache, edema, non-effective anti-inflammatory, intense redness, severe cramps, diarrhea, inflammation, symptoms for more than seven days, and mucus.
Nyazema N et al. (2007) [86]	59 PH, 73 PS	PD: 31.0%	Amoxicillin (77%), cotrimoxazole (60%), erythromycin (30%), doxycycline (48%)	
Caamaño F et al. (2005) [87]	123 PH, 164 Ph	65.9%	Clamoxyl^®^ (Amoxicillin)	
Caamano Isorna F. et al. (2004) [89]	123 PH, 164 Ph	65.9%	Clamoxyl^®^ (Amoxicillin)	
Chalker J et al. (2002) [91]	44 PH (22 control and 22 intervention)	51.0% Intervention: 57.0%, n = 12.5 and Control: 45.0%, n = 10	Cefalexin (Intervention: 57% and Control: 45%)	Simple URTI in a child < 5 years old with a mild cough.

Sample size: PH, Pharmacies; Ph, Pharmacists; PS, Pharmacy Staff; INT, Interactions; Frequency of antibiotic dispensation without a prescription: PD: Percentage corresponding to pharmacists who report dispensing antibiotics without a prescription calculated using the strategy: 100% less than the percentage of pharmacists who report never dispensing antibiotics without a prescription; PSD: Percentage corresponding to pharmacists who report dispensing antibiotics without a prescription sometimes/occasionally.; PD´: Percentage corresponding to pharmacists who report dispensing antibiotics without a prescription calculated using the strategy: 100% less than the percentage of pharmacists who report never/rarely dispensing antibiotics without a prescription; Types of disease/symptoms most commonly associated with dispensation without a prescription: UTI, Urinary Tract Infection; URTI, Upper Respiratory Tract Infection; STD, Sexually Transmitted Diseases. ^a^ Articles that do not explicitly mention throughout the full article that it is about dispensing antibiotics without a prescription, but rather the management of diseases/symptoms in pharmacies/drugstores.

**Table 4 antibiotics-09-00786-t004:** Questions and advice provided when dispensing antibiotics without a prescription.

Authors (Year)	Questions Asked at the Time of Dispensation	Advice Given at the Time of Dispensation
**Simulated patient method**		
Al-Tannir M. et al. (2020) [17]	**2011** In all scenarios presented (sore throat, acute sinusitis, otitis media, acute bronchitis, diarrhea, and UTI), none of the pharmacies asked about the drug allergy history	**2011** In all scenarios presented (sore throat, acute sinusitis, otitis media, acute bronchitis, diarrhea, and UTI), none of the pharmacies provided information on potential drug–drug interactions
	**2018** Asked about history of drug allergies (n = 4, 9.8%)	**2018** Provided information regarding potential drug-drug interactions (n = 21, 51.2%)
Halboup A. et al. (2020) [30]	**Sore throat** Asked whether the woman was pregnant (n = 31, 15.6%) **Cough** Asked whether the woman was pregnant (n = 20, 10.8%); Asked about the type of cough (productive or dry) (n = 166, 83.0%). **Diarrhea** Asked whether the woman was pregnant (=11, 7.3%) **Otitis media** Asked whether the woman was pregnant (=3, 2.0%)	**Sore throat** Explained how to use antibiotics (n = 172, 86.4%) Educated the patient about treatment duration (n = 145, 72.9%) **Cough** Explained how to use antibiotics (n = 72, 39.1%) Educated the patient about treatment duration (n = 12, 6.5%) **Diarrhea** Explained how to use antibiotics (n = 117, 75.9%) Educated the patient about treatment duration (n = 78, 50.6%) **Otitis media** Explained how to use antibiotics (n = 98, 95.1%) Educated the patient about treatment duration (n = 97, 95.1%) **UTI** Explained how to use antibiotics (n = 85, 88.5%) Educated the patient about treatment duration (n = 90, 61.2%)
Alrasheedy AA. et al. (2019) [34]	______	**Pharyngitis** Education and counseling about the importance of adherence and appropriate use of antibiotics (50.0%) **UTI** Education and counseling about the importance of adherence and appropriate use of antibiotics (52.0%)
Chang J. et al. (2019) [7]	**Diarrhoea** Asked about drug allergy history (n = 188, 16.1%) **URTI** Asked about drug allergy history (n = 494, 29.2%)	**Diarrhoea** Provided medication advice (n = 251, 21.5%) **URTI** Provided medication advice (n = 403, 23.8%)
Damisie G et al. (2019) [35]	______	**Sore throat** Explained how to take the antibiotics (n = 9, 64.3%) Explained how to take the antibiotics and duration of treatment (n = 1, 7.1%) Explained instruction on side effects (n = 1, 7.1%) Providing no counseling (n = 3, 21.4%) **Acute diarrhea** Explained how to take the antibiotics (n = 5, 31.2%) Explained how to take the antibiotics and duration of treatment (n = 4, 25.0%) Explained instruction on side effects (n = 1, 6.2%) Provided no counseling (n = 3, 18.7%) **UTI** Explained how to take the antibiotics (n = 15, 88.2%) Explained instruction on side effects (n = 1, 5.9%) Provided no counseling (n = 1, 5.9%)
Koji EM et al. (2019) [37]	Asked about drug allergy history (n = 19); Asked whether a doctor’s visit had taken place (n = 104); Asked about child’s symptoms (n = 70)	______
Mengistu G et al. (2019) [11]	None of the pharmacists asked about medication history and nutrition condition	None of the pharmacists provided infomed information about side effects and major interactions
Zawahir S et al. (2019) [38]	Further questioned about their symptoms or concurrent medical conditions (n = 36, 36.0%) Questions related to action that has already been taken (n = 12, 12.0%) Questions related to drug allergies (n = 10, 10.0%) Questions related to concurrent medicines used (n = 2, 2.0%)	In 18.0% (n = 44) of the instances, pseudo patients were recommended to see a physician, in about a quarter of them (n = 11, 25.0%) an antibiotic was still provided Explained how to take (n = 59, 60.0%) Explained how often to take (n = 47, 47.0%) Explained when to stop taking (n = 22, 22.0%)
Erku D.A. et al. (2018) [43]	Asked about drug allergies (n = 7, 8.1%) Queries about past medical and medication history (n = 18, 20.9%)	Instruction on dose and duration (n = 36, 41.9%) Instruction on side effects (n = 24, 27.9%) Advice to visit physician (n = 9, 10.6%) Non-pharmacological advice (n = 12, 14.0%)
Horumpende PG et al. (2018) [20]	______	None of the pharmacies/retailers voluntarily explained the possible side effects
Chang J. et al. (2017) [52]	**Paediatric diarrhoea** Asked further questions about the patient’s condition (n = 58, 40.6%) Asked about drug allergies (=85, 59.4%) Enquired about other symptoms (n = 6, 4.2%) Asked whether the patient had taken other drugs (n = 3, 2.1%) **Adult acute URTI** Further enquired regarding patient’s condition (n = 160, 80.4%) Asked whether had other symptoms or not (n = 64, 32.2%) Asked whether had taken other drugs or not (n = 13, 6.5%) Asked about the drug allergy history (n = 82, 41.2%)	**Paediatric diarrhoea** Provided medication advice (n = 25, 17.5%) **Adult acute URTI** Provided medication advice (n = 19, 9.6%)
Marković-Peković V et al. (2017) [18]	**2010** Patient information given (Written) (n = 59, 77.6%) Patient information given (Oral) (n = 72, 94.7%) Patient information given (Both) (n = 57, 75.0%) Patient information given (None) (n = 2, 2.6%) Asked about penicillin allergy (n = 59, 77.6%) Asked about taking other medicines (n = 19, 25.0%)	______
	**2015** Patient information given (Written) (n = 36, 50.7%) Patient information given (Oral) (n = 46, 64.8%) Patient information given (Both) (n = 32, 45.1%) Patient information given (None) (n = 21, 29.6%) Asked about penicillin allergy (n = 45, 64.3%) Asked about taking other medicines (n = 16, 22.5%)	______
Okuyan B. et al. (2017) [54]	None of the pharmacists asked about drug allergies	None of the pharmacists provided any information about other medications that could be used if an unusual condition occurred or if the patient forgot to take the medication
Guinovart MC et al. (2016) [57]	In 88 cases (73.9%) the patient was not asked about background of allergies to any antibiotics In none of the cases was the patient asked if she was pregnant Asked whether the patient was taking contraceptive treatment (n = 2, 1.7%)	Advice to visit a physician (36.1%) Explained the duration and treatment (n = 114, 95.8%)
Almaaytah A et al. (2015) [62]	Asked about drug allergy (n = 26, 17.3%) Asked about the concomitant use of other drugs (n = 8, 5.3%)	Explained how to take the antibiotic (n = 143, 95.3%) Explained the duration of treatment (n = 25, 16.7%) Recommended consulting a physician (n = 6, 4.0%)
Shet A et al. (2015) [65]	None of the pharmacists asked about drug allergies	None of the pharmacies provided counseling on expected side effects Instructions regarding the dose of the antimicrobial drugs (n = 101, 58.0%) Instructions regarding the duration of the antimicrobial drugs (n = 89, 51.1%)
Alabid A.H.M.A. et al. (2014) [67]	Asked “What symptoms have you got?” (n = 29) Asked “How long have you had the symptoms?” (n = 21) Concerning soliciting information about the colour of sputum (n = 11 (23.9%) Asked “Is there any blood in sputum?” (n = 2) Asked “How many times/year you presented the same complaint?” (n = 1) What medicines have you used before for? (n = 4) Concerning soliciting information about allergies to medicines (n = 14)	______
Marković-Peković V et al. (2012) [74]	______	Instructions for use given to the patients were oral (95%), written (78%), both (75.0%), and none (3.0%)
Rathnakar U.P. et al. (2012) [75]	None of the pharmacists asked about drug allergies	Frequency advised without asking (n = 18) Frequency advised after asking (n = 13) Duration advised without asking (n = 19) Duration advised after asking (n = 12)
Simó S et al. (2012) [76]	None of the pharmacies asked about drug allergies	None of the pharmacies explained the adverse effects
Puspitasari HP et al. (2011) [79]	In all the scenarios presented (product request for ciprofloxacin 10 tablets 500 mg; product request for 2 capsules tetracycline 250 mg and amoxicillin dry syrups 125 mg per 5 mL), none of the respondents asked about allergies In 2 of 3 scenarios (product request for ciprofloxacin 10 tablets 500 mg and product request for 2 capsules tetracycline 250 mg), none of the respondents asked about other medications taken by the patient	In the scenarios (product request for ciprofloxacin 10 tablets 500 mg and amoxicillin dry syrups 125 mg per 5 mL), none of the pharmacists informed about side effects, precautions/interactions/contra-indications and the risks of the medicine if not taken
Hadi U et al. (2010) [80]	The patients were never questioned or referred to a physician	______
Plachouras D et al. (2010) [82]	No comment was made by the pharmacist and no reason for the intended antibiotic use was requested (n = 107, 85.0%)	In the Amoxicillin + clavulanic acid case: n = 3 (4.2%) cases of dispensing, the collaborator was informed by the pharmacist about adverse events or asked whether such events had occurred in the past when the buyer had used antibiotics
Llor C et al. (2009) [83]	Asked patient about other symptoms (n = 61, 68.5%) Asked about drug allergies (n = 15, 16.9%) Asked patient whether she might be pregnant (this question was only necessary in the cases of UTI because the other clinical cases were presented by men) (n = 2, 3.6%)	Explained how often to take the antibiotic (n = 74, 83.1%) Explained how long the antibiotic should be taken (n = 62, 69.7%) Recommended that patient should see a physician if there was not any improvement (n = 4, 4.5%)
Wachter DA et al. (1999) [93]	In both scenarios, none of the pharmacies asked about drug allergies In both scenarios, none of the pharmacies asked about pre-existing medical conditions	______
**Pharmacy interview/questionnaire method**		
Hallit S et al. (2019) [36]	Age (80.1%) Weight (80.7%)	Instructed the parents to shake the bottle before each administration (81.2%) Dilute the antibiotic to the indicated line (37.1%) Store the antibiotic in the refrigerator (64.4%) Give the exact dose (53.0%) Administer it on time (46.5%) and for a defined duration of treatment (47.5%) Do not stop the antibiotic before consulting a physician or pharmacist (32.2%)
Kalungia AC et al. (2016) [58]	Asked the indication for using the specific antibiotic requested (94.0%)	Counselled on dosage instructions (n = 70, 95.9%) Counselled on common side effects (n = 22, 30.1%) No advice (n = 3, 4.1%) Were involved in suggesting changes to the antibiotic choice or brand (97.0%)
Bahnassi A. (2015) [63]	Asking for the antibiotic indication (36.0%)	No counseling (66.0%) Dosing directions (34.0%)
Bahnassi A. (2014) [68]	Provided an antibiotic without asking for the indication (36.0%)	Finish the antibiotic even when symptoms are relieved (28.0%) Discussed interactions with other medications (32.0%) Discussed possible adverse reactions (58.0%) Discussed dose and dosing regimens (82.0%)
Gastelurrutia M.A. et al. (2014) [70]	______	Just dispensed (56.9%) The doctor referred (12.1%)
Sabry NA et al. (2014) [71]	**Upon pharmacist´s recommendation**: None of the pharmacies asked about drug allergies **Upon patient´s request**: None of the pharmacies asked about drug allergies	**Upon pharmacist´s recommendation**: The pharmacist provided advice and usage instructions to 124 patients (77.5%) **Upon the patient´s request**: The dispensing pharmacist advised the patient to see the doctor (n = 8)

UTI, Urinary Tract Infection; URTI, Upper Respiratory Tract Infection.

**Table 5 antibiotics-09-00786-t005:** Comparison of the frequency of antibiotic dispensation without a prescription between the two methods.

Country	Simulated Patient Method	Pharmacy Interview/Questionnaire Method
Saudi Arabia	Al-Tannir M. et al. (2020) [17]: [2011: 77.6%; 2018: 12.5%] Alrasheedy AA. et al. (2019) [34]: 92.2% Al-Mohamadi A et al. (2011) [78]: 97.9% Al-Ghamdi MS. (2001) [92]: 82.0%	Alrasheedy AA. et al. (2019) [34]: 70.7% Alhomoud F et al. (2018) [41]: 100.0% Bahnassi A. (2014) [68]: 100.0%
Yemen	Halboup A. et al. (2020) [30]: 73.3%	Abood EA et al. (2016) [55]: 25.3%
Egypt	Abdelaziz AI et al. (2019) [33]: 98.4%	Sabry NA et al. (2014) [71]: 18.2%
Ethiopia	Damisie G et al. (2019) [35]: 94.4% Koji EM et al. (2019) [37]: 63.4% Mengistu G et al. (2019) [11]: 86.7% Erku D.A. et al. (2018) [43]: 86.0% Abegaz T.M. et al. (2016) [14]: 51.3%	Mengistu G et al. (2019) [11]: 50.5% Erku D.A. (2016) [56]: 90.2%
India	Nafade V. et al. (2019) [19]: 4.0% Satyanarayana S. et al. (2016) [61]: 27.0% Shet A et al. (2015) [65]: 66.7% Rathnakar U.P. et al. (2012) [75]: 51.7%	Barker A.K. et al. (2017) [51]: 100.0% Shreya Svitlana A. et al. (2015) [66]: 100.0%
Sri Lanka	Zawahir S et al. (2019) [38]: 41.0% Zawahir S et al. (2018) [49]: 61.0% Wolffers I. (1987) [94]: 100.0%	Zawahir S. et al. (2019) [39]: 31.7%
Indonesia	Puspitasari HP et al. (2011) [79]: 91.0% Hadi U et al. (2010) [80]: 75.9%	Ajie A.A.D. et al. (2018) [40]: 92.1%
Pakistan	Malik M. et al. (2013) [15]: 28.57%	Rehman IU et al. (2018) [46]: 68.0% Sarwar M.R. et al. (2018) [47]: 93.7% Nawab A. et al. (2016) [60]: 12.2%
Spain	Zapata-Cachafeiro M et al. (2018) [48]: 18.8% Guinovart MC et al. (2016) [57]: 54.1% Simó S et al. (2012) [76]: 8.0% Llor C et al. (2010) [81]: 45.2% Llor C et al. (2009) [83]: 45.2%	Gastelurrutia M.A. et al. (2014) [70]: 9.8% Zapata-Cachafeiro M. et al. (2014) [72]: 64.7% Caamaño F et al. (2005) [87]: 65.9% Caamano Isorna F. et al. (2004) [89]: 65.9%
Nepal	Wachter DA et al. (1999) [93]: 67.5%	Ansari M. (2017) [50]: 66.5%
Syria	Al-Faham Z et al. (2011) [77]: 97.0%	Mansour O. et al. (2017) [53]: 85.5% Bahnassi A. (2015) [63]: 100.0%
Malaysia	Alabid A.H.M.A. et al. (2014) [67]: 32.0%	Khan M.U. et al. (2016) [59]: 63.3%
Brazil	Volpato DE et al. (2005) [88]: 74.0%	Rauber C. et al. (2009) [84]: 85.0%
Vietnam	Chalker J et al. (2000) [16]: 81.5%	Chalker J et al. (2002) [91]: 51.0%
Zimbabwe	Nyazema N et al. (2007) [86]: 8.0%	Nyazema N et al. (2007) [86]: 31.0%

## 3. Discussion

The dispensing of antibiotics without a prescription remains a common practice worldwide, especially in low- and middle-income countries, and the community pharmacies/drugstores continue to be a major source of antibiotic acquisition without a medical prescription [95,96]. With this is mind, there is an urgent need to improve prescription practices in low- and middle-income countries, starting from the integration of treatment recommendations into their national guidelines, and reinforcement of the awareness of this public health problem, togeher with their possible consequences to humanity. In high income countries this is a less common practice as several factors may affect prescribing behaviors, such as socio-cultural context, financial incentives, personal beliefs, patients’ attitudes and greater awareness of the antibiotic resistance problem due to an easier information access when compared to low- and middle-income countries [97,98]. The development of educational strategies in high-income countries has proven to be not enough, since the sale of antibiotics without prescription continues to emerge. So, there is a need to intensify these strategies, through the involvement of all stakeholders, with more advertising campaigns, more workshops, and more publicity. Likewise, law enforcement to reduce non-prescription sales of antibiotics with significant penalties and revocation of professional licenses in case of non-compliance also seems to be a good strategy to be adopted [99,100].

To the best of our knowledge, this is the first study that compares the frequency of dispensing antibiotics without a prescription by using two different data collection methods. Most of the studies included in this review used the simulated patient method. This may be related to the fact that the interview/questionnaire method implies greater participation and proximity to the pharmacy professional, with the lack of anonymity potentially leading to less honest answers. On the other hand, the simulated patient is an effective data collection method which provides internal validity [101].

The analysis of the results obtained in the systematic review, reveal that the values for this practice remain high, reaching 100% in some of the studies. Since the average value of the percentage of antibiotic dispensation without a prescription in the simulated patient method is very similar to the percentage found through the method with interviews/questionnaires, it seems that the Hawthorne effect did not have much impact in the studies using the first method. Furthermore, in the simulated patient method, most antibiotics were easily dispensed at low insistence levels by patients. A study conducted in 2018 in European and Anglo-Saxon countries showed that, although the consumption of antibiotics is decreasing, patients still see antibiotics as available products that can be easily bought [102].

Results of this study highlight that the highest percentages of antibiotic dispensation without a prescription occurred in Asia. The overuse of antibiotics is an evident concern in this part of the world, with these drugs being often available for sale without a prescription [103]. A large number of studies analyzed in this review took place in Asia (51 articles representing 60% of the total), which may influence the fact that they also display the highest percentage of antibiotics dispensed without prescription, since they stand out more in relation to studies carried out in other parts of the world. Our results reveal that there were few studies in Anglo Saxon countries. Thus, we alert researchers to develop studies in these countries.

Another important aspect to highlight is that in some of the articles it was possible to observe a decrease in the frequency of dispensing antibiotics without a prescription, which led us to think that the growing approach taken worldwide on this public health problem is beginning to have an impact on the practices of health professionals.

The most commonly non-prescribed antibiotics sold identified in our review were amoxicillin (53 articles), azithromycin (24 articles), ciprofloxacin (24 articles), and amoxicillin-clavulanic acid (24 articles). These findings are in agreement with results found in a previous study that identified as antibiotics mostly used for self-medication the penicillins, macrolides, cephalosporins, fluoroquinolones, and tetracycline [96]. These specific classes of drugs have been associated for some time with increased development of resistance to microorganisms such as Streptococcus pneumoniae, Acinetobacter baumannii, Campylobacter jejuni, Enterococcus faecalis, and many others [104].

The pathologies/symptoms most commonly associated with dispensing antibiotics without a prescription were respiratory system problems like sore throat, URTI, common cold, bronchitis, cough, cold and flu, diarrhea, and UTI. Previously, another study concluded that the main pathologies associated with the consumption of antibiotics without a prescription were sore throat, fever, and respiratory problems, such as cold/flu and cough [96]. Patients should be educated and encouraged by pharmacists to go to primary health care units whenever they have these minor illnesses.

A systematic review recently carried out identified the main intrinsic factors that influence the dispensing of antibiotics without a prescription, such as the low level of training of professionals working in pharmacies, as well as their attitudes such as complacency, ignorance, responsibility, indifference, and economic benefit [105]. In our study, concerning questions and advice provided by pharmacists at the time of dispensing antibiotics without a prescription showed very low percentages, with the advice being absent a few times. When this occurred, the information primarly provided corresponded to explanations about how to use antibiotics and the duration of treatment. Pharmacy professionals have an essential role in dispensing antibiotics, so interventions must be implemented with them. Most community pharmacists are aware that the irrational use of antibiotics is one of the main causes of increased antibiotic resistance and that all healthcare professionals, including themselves, need to think rationally and stop prescribing and dispensing so many antibiotics inadequately [95]. However, a previous study has found that the implementation of multiple activities among pharmacists, like national campaigns and programs, workshops for health professionals, creation of posters and flyers, television programs, newspapers and lauch of guidelines for counseling patients, improved antibiotic use and consequently reduced inappropriate antibiotic prescription and dispensation [18]. We can see that there are discrepancies between knowledge and practice.

One strengh of this review is the heterogeneous composition of the included studies that were analyzed, which increased the generalization of the results and, consequently, the external validation of this review.

A limitation of our study is that it does not take into account any intervention implemented to decrease the dispensation of antibiotics without a prescription in the countries, after the studies were carried out. Another limitation of this review concerns with the lack of inclusion of studies concerning veterinary use. A large portion of all antibiotic use is for veterinary use. Additionally, an increase in global consumption is expected to occur in livestock between 2010 and 2030, thus leading to a rise in the incidence of bacteria resistant to antimicrobials in animals, with a possible high transmission to humans [106,107].

## 4. Materials and Methods

### 4.1. Search Strategy/Search Methods for Identification of Studies

This systematic review followed the PRISMA (Preferred Reporting Items for Systematic Reviews and Meta-Analyses) guidelines, which is a template developed to help authors improving the reporting of systematic reviews and meta-analyses [108], and was registered in the PROSPERO network (registration number: CRD42020189331).

The databases MEDLINE Pubmed and Embase were searched in June 2020 using the query: “(antibiotic OR antibiotics OR antimicrobial OR antimicrobials) AND (over-the-counter OR nonprescription OR without-prescription OR self-medication) AND (community-pharmacy OR community-pharmacies OR pharmacy OR pharmacies OR pharmacist OR pharmacists OR community-pharmacist OR community-pharmacists)”, to identify studies that reported the dispensing of non-prescribed antibiotics in community pharmacies and drugstores that sell drugs for human use.

### 4.2. Study Inclusion Criteria

Studies were considered eligible for this review if they met the following criteria: (i) studies published in English, Portuguese, or Spanish; (ii) the study population was defined as pharmacies or drugstores that sell drugs for human use; (iii) studies whose data collection method consists of the simulated patient method and/or questionnaires/interviews with pharmacies; (iv) studies with original/primary data, namely observational studies, and interventional studies since they present pre-intervention/baseline data and (v) data referring to antibiotics for systemic use. Population-based studies that explored public opinion and attitudes on the sources and use of non-prescription antibiotics were excluded, as well as qualitative studies that did not present quantitative measures of non-prescription supply of antibiotics. Studies focused on the administration of antiretroviral, antimalarial, antifungal, and antiparasitic drugs were also excluded.

In articles addressing other target populations, only data referring to studies carried out in pharmacies and with pharmacy workers were extracted.

All articles extracted were independently reviewed by two authors, who decided whether or not these met the selection criteria. In case of disagreement, the paper in question was examined by a third and fourth reviewer who took the final decision.

### 4.3. Quality Assessment

All selected studies were assessed for quality and risk of bias through the Appraisal tool for Cross-Sectional Studies (AXIS). This tool consists of a 20 items questionnaire that addressed study quality and reporting, assessing whether the published conclusions are reliable and credible about the objective, methods, and results [109]. For each study, the risk of bias and quality was conducted by two researchers and, in case of disagreement, a third reviewer acted as a referee to reach a consensus.

### 4.4. Data Extraction

To summarise the general characteristics of the studies selected in the previous analysis, Table 1 was drawn up showing the following parameters: author (year of publication), study location/country, study design, and data collection method.

Additionally, to indicate the extent of antibiotic dispensation without a prescription and additional outcomes, two tables were created, one referring to the simulated patient method and the other referring to the interview/questionnaire method in pharmacies. The table related to the simulated patient method (Table 2) presents the following parameters: sample size, frequency/prevalence of antibiotic dispensing without a prescription, name/class of antibiotics most dispensed, types of diseases/symptoms most commonly associated with dispensing without a prescription and level of insistence by simulated patients.

Regarding the outcome “level of insistence by simulated patients”, the levels of insistence recorded are those defined by the authors of each study. These have been recorded in Table 2, specifically in the associated column, before the percentage for each level of insistence is presented. Regardless of the divergence in the definition of the levels by the various authors, the common issue between them is that the lower level is associated with a lower level of insistence by the simulated patient of generally only asking for medication for symptom relief. As the level increases, the simulated patients start asking for stronger medication and even specifically for antibiotics.

The pharmacy interview/questionnaire method table (Table 3) shows the following outcomes: sample size, frequency/prevalence of antibiotic dispensation without a prescription, name/class of antibiotics most often dispensed and types of diseases/symptoms most commonly associated with dispensation without a prescription.

Another table (Table 4) was constructed to record the addressed questions, the advice given at the time of antibiotics dispensing without a prescription, as well as the percentage of pharmacies/pharmacists who performed them.

Subsequently, data on the comparison of the frequency of dispensing antibiotics without a prescription between the two methods were extracted and introduced in Table 5.

Data were extracted by two reviewers and any discrepancy was resolved by a third and fourth reviewer.

## 5. Conclusions

The dispensing of antibiotics without prescription is one of the major drivers of antibacterial resistance. Our results suggest that this practice is still common in many countries, especially in low- and middle-income countries. Pharmacies and their work professionals play a critical role in the conservation of effective antibiotics, through denying the dispensing of antibiotics without a prescription and improving patient pharmaceutical counseling, overall in respiratory and UTI and in diarrhea. It becomes urgent to empower these health professionals, especially in developing countries, through the implementation of educational and/or administrative strategies in order to reduce the dispensation of antibiotics without a prescription.

## Figures and Tables

**Figure 1 antibiotics-09-00786-f001:**
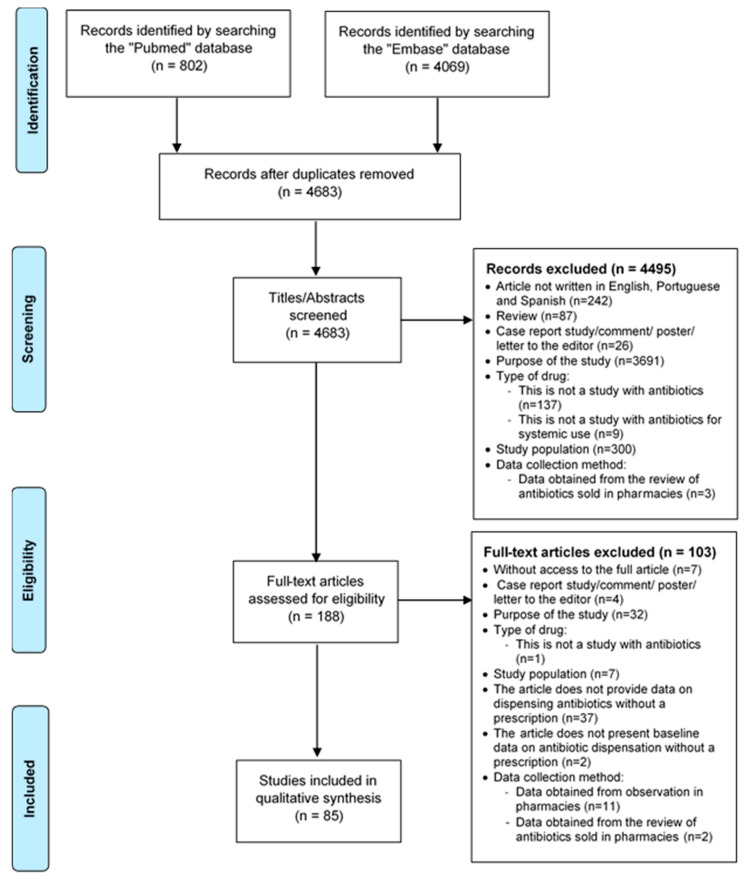
Study selection flowchart.

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
