# Peer review of "Antibiotic Dispensation without a Prescription Worldwide: A Systematic Review"

_antibiotics, 2020, doi:10.3390/antibiotics9110786_

Round 1

Reviewer 1 Report

This systematic review gives an overview of antibiotic dispensing practices in pharmacies in different countries in the world.

While the topic certainly is of interest, the execution of the idea lacks substance. The authors generalize too much and their claims are not justified on a global scale.

Some comments below:

  1. Abstract: does not give enough details on countries included, antibiotic classes and indications. There is no justified reasoning for the last sentence. Abstract too vague.
  2. Line 40: this is based on 1 article with data from 2000-2015, there are multiple countries where antibiotic prescriptions have decreased in recent years
  3. Line 44/45: “It is estimated that over-the-counter dispensing may account for half of antibiotic sales worldwide, even though in North America, Northern Europe, and Australia it is practically non-existent.” Correct. That is why the authors should not generalize their review outcomes on a global scale!
  4. Lines 50-55: rephrase, unclear, elaborate more
  5. Lines 109-114: therefore, no generalization on a global scale
  6. Line 120: difference between class I and II pharmacies has not been explained
  7. Lines 125-128: explanations and definitions are missing
  8. A summary of the AXIS tool for quality assessment results of the included articles would be useful in table 1, so the authors can show the quality of each individual paper included in the review.
  9. Line 139: define the levels of demand in the review before you discuss them.
  10. Line 265: I do not understand the reasoning here: Why would the simulated patient method being published 4 years after the interview/questionnaire paper make you to expect a decrease in the frequency of antibiotic dispensation without a prescription?
  11. Line 332: this article is from 2010, too old to support this claim!
  12. Line 333: it would be advisable to re-write the entire article with a focus only on Asia
  13. Line 435: this sounds like bad parenting. Unprofessional writing. What different ways and examples are to improve the situation? What kind of educational strategies? In the current form the conclusions are too vague.
  14. There are two Table 2’s. Fix table numbers and reference to them in the text.
  15. Table 3 - There are two issues here: (1) Unweighted averaging of simulated method papers vs pharmacy interview papers is inappropriate. If you are going to average all the antibiotic dispensation rates in each nation, the results need to be weighted for their sample size contribution when taking the global average, otherwise smaller studies are being given more weight in influencing the average. (2) The difference in averages you are stating are differences in percentage points, not a percentage difference. For example, for Saudi Arabia the difference is 17.8 percentage points, not 17.8% different. Change the heading to “Difference (percentage points)”.
  16. The English language needs to be improved. There are many grammar errors, spelling mistakes and informal writing throughout the manuscript, inconsistencies in font, font size,… There are several sentences starting with a number.

Reviewer 2 Report

Congratulations for such an important piece of work! The following comments are made towards further improving this work.

There are some minor issues with the written English that should be revised.

Line 28 - “50 articles present”  - suggest “presenting”

Line 51 – “pharmacists studied”. Suggestion “studied pharmacists”

Line 104 – “average amount of dispensation”. I would suggest “the average dispensing rate…”

Line 112 – “the oldest year to the most recent”. I would suggest “from the past to more recent years”

Line 121 – a sentence starts with “2”. It would be better starting with “Two”

Line 369 – “One strenghness”. Suggest revising to “strength”

Line 370 – add “which increases”.

Background

Consumption of antibiotics could be for human or animal use, with the latter accounting for the biggest percentage of consumption. To the objective of this study, it would be important to distinguish between human and animal consumption of antibiotics.

Results:

Line 120 – please mention the countries

Line 151 – Table 1 title should be “Characteristics of the selected studies”

Please review table numbers (there are two table 2), and the formatting of each table (letter, spacing, etc.).

Discussion:

I was expecting more in this section.

First, you state that this a common practice worldwide. But as per your review, apparently the evidence suggests that this is mainly a low and middle income countries’ (LMIC) problem. The authors reinforce this view in the background section, stating that “in North America, Northern Europe, and Australia selling of antibiotics without prescription is practically non-existent”. But, specifically for Europe, if 50% of patients obtain antibiotics without prescription in Europe (evidence stated in line 331), which hints at some lack of research effort of these countries on this issue. This is evident from this review, with very few studies and most coming from Spain. So, something is missing, and maybe these countries have no interest in researching this practice or researchers in LMIC acknowledge that the problem still exists, and try to study the phenomenon. What can be done about it? What other hypothesis could you offer that could help other researchers in the field to design and implement studies in high income countries to address the issue of antibiotic dispensing without prescription?

The authors hint that education strategies are needed. Since this issue is addressed for a long time in most countries, where educational strategies are already in place, one wonders if these are enough. As by the example of the country where two types of pharmacy exists suggest, this is also a policy problem. The education strategies have not been enough.

What other strategies could there be? What strategies North America, Northern Europe, and Australia follow that go beyond the educational intervention? More on this would improve your discussion.

Line 366 – What multiple activities? Examples are needed.

Conclusions:

Conclusions should reflect more on the results of the study, instead of just presenting some ideas from the discussion.

Reviewer 3 Report

This is a very interesting work from Batista AD et al, about a very hot and important topic in our time of struggle against antimicrobial resistance. The article has a strong and accurate methodology providing data about those countries where there is apparently the most urgent need to implement interventions to tackle antimicrobial resistance.However I suggest a revision of the English grammar with some minor corrections:

Abstract Line 22: “importantdriver”, of course is a typing error

Manuscript

Line 106: “Finally, only the article Ibrahim M.I.B.”, I would suggest to add “only the article by Ibrahim M.I.B.”

Line 109: “Two of the articles included in the review, Al-Tannir M et al (2020) and Marković-Peković V. et al”, I would add “those by Al-Tannir M et al (2020) and Marković-Peković V. et al”.

Line 119: “3 articles” should be changed into “Three articles”

Line 121: “2 of this article” should be changed into “Two of these articles”

Line 125: “Additionally, the article Minzi O et al.”, I would add A”dditionally, the article by Minzi O et al.”

Authors sometimes put references to cite the different studies, other times they write author and year of publishing, I suggest citing always the reference in order to help the readers in following the developing of the manuscript.

Table 1

References are missing for three cited studies:

  • “Mohamed Ibrahim M.I, et” 2018
  • “Marković-Peković V. et al” 2017 and 2012

Line 159: “7 articles” should be changed into “Seven articles”

Line 162: “The article Hadi U et al. (2010)”, please add reference and “The article by Hadi U et al”

Line 164: “8 articles” should be changed into “Eight articles”

Line 169: “Al-Tannir M. et al. (2020)”, please add reference and “The article by Al-Tannir M.  et al”

Line 184: “4 studies present a percentage of 100%” should be changed into “Four studies…”

Line 196: “In the article Paes M.R. et al. (2018)” please add reference and “The article by Paes M.R et al”

Line 197: “in the article Nawab A. et al. (2016)” please add reference and “The article by Nawab A. et al”

Line 217: “2 articles” should be changed into “Two articles”

Lines 218 and Line 219: “in the article Kalungia AC et al. (2016) and the value of 58% in the article Bahnassi A. (2014)” please add reference and change into “in the article by Kalungia AC et al. (ref) and the value of 58% in the article by Bahnassi A. (ref)”

Lines 253 through 257: “… from the Al-Ghamdi MS study published in 2001 (82%), for the Al-Mohamadi A et al. published in 2011 (97.9%) and afterwards it is possible to observe a slight decrease, with the study Alrasheedy AA. et al. (2019) registered a value  of around 92.15%. The study Al-Tannir M. et al. published in 2020 it registered the lowest value both in the year of 2011 (77.6%) and in the year of 2018 (12.5%).” Please put references instead of year of publishing and change the sentences as following “… from 82% in the study by Al-Ghamdi MS et al (ref), and 97.9% by Al-Mohamadi A et al (ref) to 92.15% in the work by Alrasheedy AA. et al (ref), observing a slight decrease. The study by Al-Tannir M. et al. (ref) registered the lowest value both in 2011 (77.6%) and in 2018 (12.5%).”

Lines 260 through 262: “from the Bahnassi A. study published in 2014 for the Alhomoud F. et al. published in 2018. Conversely, there was a decrease in this percentage in the Alrasheedy AA. et al. study published in 2019 (70.7%).” Please put references and change sentences into “from the study from Bahnassi A. et al (ref) and the one from Alhomoud F. et al. (ref). Conversely, there was a decrease in this percentage up to 70.7% in the study by Alrasheedy AA. et al. (ref)”

Lines 279 through 282: “with the article Alrasheedy AA. et al. (2019) carried out in Saudi Arabia registered a difference between the two methods of 21.45% and the article  Mengistu G et al. (2019) conducted in Ethiopia a difference between the two methods of 36.2%. Alternatively, the article Nyazema N et al. (2007)”, please put references and change sentences into “the article by Alrasheedy AA. et al. (ref) carried  out in Saudi Arabia registered a difference between the two methods of 21.45%, and the article by Mengistu G et al. (ref) conducted in Ethiopia a difference between the two methods of 36.2%.  Alternatively, the article by Nyazema N et al. (ref)”

Table 2

  • Shi L. et al. (2020)[28] “Three levels of demand: level 1 (client required some medicine for cough) (22.5%), level 2 (cliente explicitly expressed the requirement of antibiotic) (60.5%), level 3 (client specifically required roxithromyci) (5.4%)”, typing error please change “cliente” into “client”

Discussion

Lines 333 and 334: “Results of this study highlight that the highest percentages of antibiotic dispensation without a prescription are associated with Asia”, I would rather mitigate the sentence changing it into “Results of this study highlight that the highest percentages of antibiotic dispensation without a prescription occurred in Asia”

Lines 335 through 337: “Besides that, most of the studies included in this review were also conducted in Asia and this also may be related to this result” please clarify this sentence

Round 2

Reviewer 1 Report

n/a